# miR-142 orchestrates a network of actin cytoskeleton regulators during megakaryopoiesis

Elik Chapnik[1†], Natalia Rivkin[1†], Alexander Mildner[2], Gilad Beck[1], Ronit Pasvolsky[1], Eyal Metzl-Raz[1], Yehudit Birger[3], Gail Amir[4], Itay Tirosh[1], Ziv Porat[5], Liron L Israel[6,7], Emmanuel Lellouche[6,8], Shulamit Michaeli[6,8], Jean-Paul M Lellouche[6,7], Shai Izraeli[3,9], Steffen Jung[2], Eran Hornstein[1*]

[1]Department of Molecular Genetics, Weizmann Institute of Science, Rehovot, Israel; [2]Department of Immunology, Weizmann Institute of Science, Rehovot, Israel; [3]Functional Genomics and Leukemic Research, Cancer Research Center, Sheba Medical Center, Ramat Gan, Israel; [4]Department of Pathology, Hadassah Medical Center, Jerusalem, Israel; [5]Department of Biological Services, Weizmann Institute of Science, Rehovot, Israel; [6]Institute of Nanotechnology and Advanced Materials, Bar-Ilan University, Ramat-Gan, Israel; [7]Department of Chemistry, Bar-Ilan University, Ramat-Gan, Israel; [8]The Mina and Everard Goodman Faculty of Life Sciences, Bar-Ilan University, Ramat-Gan, Israel; [9]Department of Human Molecular Genetics and Biochemistry, Tel Aviv University, Tel Aviv, Israel

*For correspondence: eran. hornstein@weizmann.ac.il

†These authors contributed equally to this work

**Abstract** Genome-encoded microRNAs (miRNAs) provide a posttranscriptional regulatory layer that controls the differentiation and function of various cellular systems, including hematopoietic cells. miR-142 is one of the most prevalently expressed miRNAs within the hematopoietic lineage. To address the *in vivo* functions of miR-142, we utilized a novel reporter and a loss-of-function mouse allele that we have recently generated. In this study, we show that miR-142 is broadly expressed in the adult hematopoietic system. Our data further reveal that miR-142 is critical for megakaryopoiesis. Genetic ablation of miR-142 caused impaired megakaryocyte maturation, inhibition of polyploidization, abnormal proplatelet formation, and thrombocytopenia. Finally, we characterized a network of miR-142-3p targets which collectively control actin filament homeostasis, thereby ensuring proper execution of actin-dependent proplatelet formation. Our study reveals a pivotal role for miR-142 activity in megakaryocyte maturation and function, and demonstrates a critical contribution of a single miRNA in orchestrating cytoskeletal dynamics and normal hemostasis.

## Introduction

microRNAs (miRNAs) are single-stranded RNA molecules of 22 nucleotides in length, processed from endogenous hairpin transcripts. miRNAs provide cells with a sequence-based silencing mechanism through base-pairing of a minimal recognition sequence, called the miRNA 'seed' (*Bartel, 2009*; *Carthew and Sontheimer, 2009*; *Fabian et al., 2010*).

miRNAs control hematopoiesis and the function of both lymphoid and myeloid progeny (*Chen and Lodish, 2005*; *Lawrie, 2007*; *Garzon and Croce, 2008*; *Navarro and Lieberman, 2010*). For example, *in vivo* studies uncovered roles for miR-451 in erythropoiesis (*Patrick et al., 2010*; *Rasmussen et al., 2010*; *Yu et al., 2010*), for miR-223 in granulopoiesis (*Johnnidis et al., 2008*), for miR-150 in the commitment of multipotent myeloerythroid progenitors (*Lu et al., 2008*), and for miR-155 in the

**eLife digest** DNA carries all the information needed for life. This includes the codes required for making proteins, as well as instructions on when, where, and how much of these proteins need to be produced. There are a number of ways by which cells control protein manufacturing, one of which is based on small RNAs called microRNAs. Before proteins are assembled, the DNA molecule is copied into a temporary replica dubbed messenger RNA. microRNAs are able to recognize specific messenger RNA molecules and block protein production.

microRNAs serve a very important regulatory role in our bodies and are involved in virtually all cellular processes, including the production of all classes of blood and immune cells. Platelets seal injuries and prevent excessive bleeding by creating a clot at the location of a wound. Platelets are produced in huge cellular factories called megakaryocytes, which need to have a flexible and dynamic internal skeleton or cytoskeleton to produce the platelets.

Chapnik et al. focus on one specific microRNA gene, which is vital for the production and the function of several classes of blood and immune cells. Chapnik et al. created a mouse model that does not produce one specific microRNA—miR-142—and found that mutant mice produced fewer platelets than normal mice. Although one possible explanation for this is that the mutant mice also had fewer megakaryocytes than normal, Chapnik et al. unexpectedly found that the number of megakaryocytes was in fact higher. However, these megakaryocytes do not reach functional maturity, which is required for platelet production. Many of the megakaryocytes made by the mutant mice were also smaller than normal and had an unusual cytoskeleton.

Using a genomic approach and molecular tools, Chapnik et al. show that miR-142 affects the production of several of the proteins responsible for the dynamic flexibility of the cytoskeleton in mature megakaryocytes. Therefore, a single microRNA can target multiple different proteins that coordinate the same pathway in the cells that are critical for clotting and hence for human health.

miR-142 has also been suggested to have important functions in blood stem cells and in blood cancer (leukemia). Therefore, the new mouse model could be used to investigate many other facets of the blood and immune system. Further research could also focus on whether the same cytoskeletal network is in charge of miR-142 activity in other blood cells, or if miR-142 silences different targets in different cells.

mammalian immune system (*Rodriguez et al., 2007*; *Thai et al., 2007*; *Mann et al., 2010*; *O'Connell et al., 2010*, *2007*). Megakaryocytes (MKs) display a distinctive miRNA expression pattern (*Garzon et al., 2006*; *Opalinska et al., 2010*; *Xu et al., 2012*). However, functional genetic studies dissecting the role of miRNA in megakaryopoiesis are still limited.

In the present work, we focused on miR-142, a hematopoietic-specific miRNA, which resides in a genomic locus that was previously associated with t(8;17) translocation in B-cell leukemia (*Gauwerky et al., 1989*). Pioneering experimental evidence has suggested miR-142 involvement in lymphocyte differentiation (*Chen et al., 2004*) and recently, miR-142 was also shown to play a role in the specification of definitive hemangioblasts (*Lu et al., 2013*; *Nimmo et al., 2013*), and in lymphoid and myeloid lineages (*Gauwerky et al., 1989*; *Chen et al., 2004*; *Huang et al., 2009*; *Belz, 2013*; *Lagrange et al., 2013*; *Lu et al., 2013*; *Nimmo et al., 2013*; *Sonda et al., 2013*; *Sun et al., 2013*; *Zhou et al., 2013*). Furthermore, miR-142 is involved in the compound immune response to North American eastern equine encephalitis virus (*Trobaugh et al., 2014*) and our work uncovered a key role for miR-142 in the maintenance of CD4[+] dendritic cells (*Mildner et al., 2013*).

MK maturation is an intricate process that involves DNA replication in the absence of cytoplasmic division, termed endomitosis. Polyploid MKs also exhibit distinctive cytoskeletal rearrangements that enable the biosynthesis of platelets (also known as thrombocytes). The elaborated actin cytoskeleton of MKs is uniquely organized in a way that allows cytoplasmic proplatelet protrusions to bend and bifurcate into multiple tips, from which platelets are subsequently released to the bloodstream. Dysregulation of the actin cytoskeleton impairs the generation of mature platelets (*Hartwig and Italiano, 2006*; *Bender et al., 2010*). Accordingly, actin-modulating genes were shown to be crucial for MK maturation and have been implicated in the etiology of platelet-related disorders (*Villa et al., 1995*; *Kajiwara et al., 1999*).

We characterize miR-142-deficient mice that display an array of hematological defects, including pronounced thrombocytopenia. We show that miR-142 controls multiple facets of MK differentiation including control of cell size, ploidy, and proplatelet elaboration. Furthermore, we demonstrate that miR-142-3p controls platelet biosynthesis by orchestrating the coordinated expression of several distinct nodes in a network of actin cytoskeleton regulators. Our study reveals a novel miRNA-dependent circuit that maintains cytoskeletal integrity, and suggests that a single miRNA may broadly regulate cell function by controlling a coherent set of effectors in a given pathway.

## Results

### miR-142 controls thrombopoiesis

A miR-142$^{-/-}$ allele was created by insertion of an exogenous gene trap sequence ~50 bp upstream of the murine pre-miR-142. This cassette disrupted normal transcription and drove the expression of a beta-galactosidase reporter gene (*Stanford et al., 2001*; *Hansen et al., 2008*; *Osokine et al., 2008*; *Figure 1A*).

To confirm miR-142 nullification, we collected circulating mononuclear cells from peripheral blood of homozygous mutant mice and wild-type (WT) littermates. miR-142-3p is the guide strand from the miR-142 hairpin, whereas the sister 'passenger' miR-142-5p strand is expressed in negligible levels (*Nimmo et al., 2013*; *Figure 1B*). The expression of both miR-142-3p and miR-142-5p was abolished in miR-142$^{-/-}$ circulating blood cells as exemplified by quantitative real-time PCR (qPCR, *Figure 1B*).

Genotyping the progeny of miR-142$^{+/-}$ intercrosses at embryonic day 14.5 (E14.5) revealed the predicted Mendelian distribution of miR-142$^{+/+}$, miR-142$^{+/-}$, and miR-142$^{-/-}$ embryos. However, postnatal survival at 3 weeks of age of miR-142 homozygous offsprings was lower than expected (18% instead of 25%), demonstrating that roughly a third of miR-142$^{-/-}$ mice died perinatally (*Figure 1—source data 1*). Surviving miR-142$^{-/-}$ mice did not display overt physical abnormalities, were fertile, and bred normally.

To comprehensively characterize miR-142 expression pattern, we performed a fluorescence-based semi-quantitative detection of beta-galactosidase activity in viable hematopoietic cells. In accordance with previous reports (*Lagos-Quintana et al., 2002*; *Chen et al., 2004*; *Ramkissoon et al., 2006*; *Merkerova et al., 2008*), this study revealed pan-hematopoietic activity of the miR-142 promoter, which drove the expression of a LacZ transgene from the endogenous miR-142 locus in all lymphoid and myeloid lineages examined (*Figure 1C*).

To assess the impact of miR-142 loss *in vivo*, we performed complete blood counts that revealed reduced numbers of erythrocytes and white blood cells in miR-142$^{-/-}$ animals relative to WT littermates (*Figure 1D,E*). Interestingly, at 2 months of age, miR-142$^{-/-}$ mice displayed a striking ~50% decrease in platelet counts and a ~10% increase in mean platelet volume (MPV), relative to controls (*Figure 1F,G*). Thus, miR-142$^{-/-}$ animals suffer from macrothrombocytopenia. The reduction in platelet numbers and the concordant increase in platelet size were even more pronounced when analyzed at 1 year of age (*Figure 1F,G*).

In addition, lethally irradiated WT mice, which were reconstituted with miR-142$^{-/-}$ bone marrow (BM) cells and analyzed 6 weeks following transfer, showed platelet paucity relative to controls that were reconstituted with WT BM cells (*Figure 1—figure supplement 1*). However, the decrease in platelet numbers in this model was less significant compared to that demonstrated in germline miR-142-deficient mice (*Figure 1F*), plausibly due to contribution of WT cells. Taken together, these data demonstrate that hematopoietic-specific miR-142 activity is required for normal platelet production.

### Perturbed myeloerythroid development in the absence of miR-142

We hypothesized that the diminished numbers of circulating platelets might stem from a defect in the development of MKs. First, we confirmed that miR-142-3p is the functional dominant 'guide' strand in MKs, and that both sister miR-142 species are nullified in miR-142$^{-/-}$ MKs (*Figure 2—figure supplement 1A*). We also confirmed that the expression of *Bzrap1*, a regulator of synaptic transmission (*Chardenot et al., 2002*; *Mittelstaedt and Schoch, 2007*) positioned 3.5 Kb downstream of miR-142, is unchanged in miR-142$^{-/-}$ MKs (*Figure 2—figure supplement 1B*).

To gain insight into the impact of miR-142 nullification on early MK development, we performed a high-resolution flow cytometry assay for the characterization of myeloerythroid progenitors (*Pronk et al., 2007*; *Pronk and Bryder, 2011*). The numbers of miR-142$^{-/-}$ bipotent MK-erythroid precursors (PreMegEs) were marginally increased relative to controls (*Figure 2A,B*).

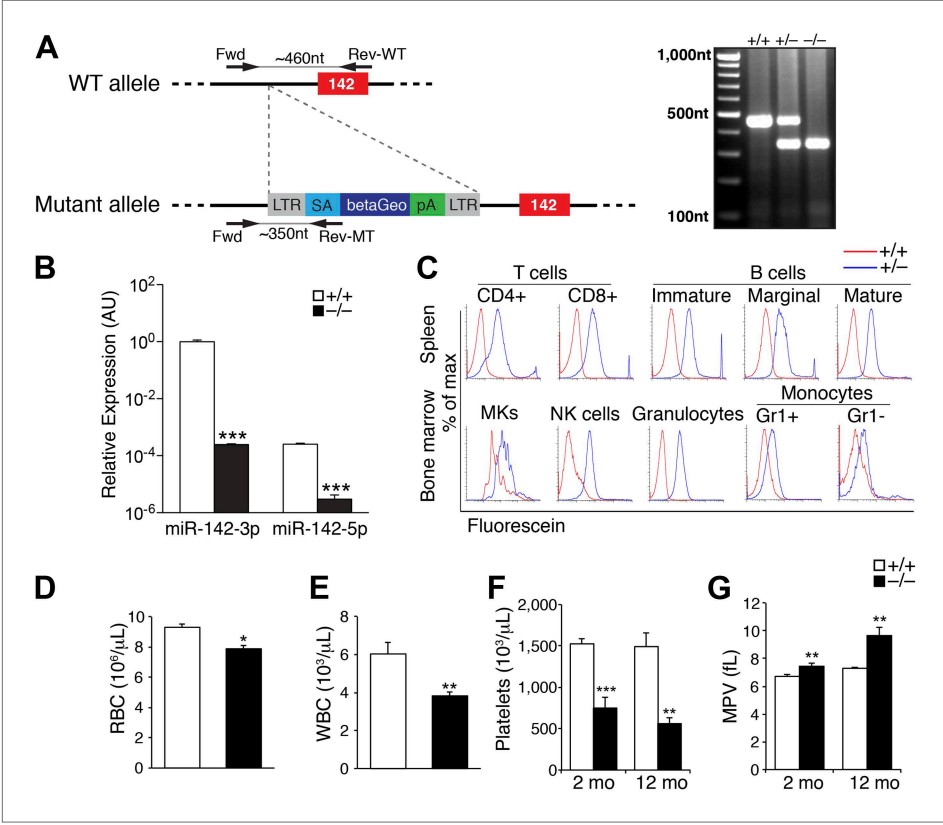

**Figure 1**. Pronounced thrombocytopenia in miR-142$^{-/-}$ mice. (**A**) Left panel: schematic representation of the gene trap cassette targeting the murine miR-142 locus. The WT and mutant loci are shown with the gene trap-targeting vector. pre-miR-142 is shown as a red box. LTR, long terminal repeats; SA, splice acceptor; betaGeo, beta-galactosidase-Neomycin resistance fusion protein; pA, polyA signal. Right panel: Genomic PCR confirmation of miR-142 trap insertion. (**B**) Quantitative real-time (q) PCR performed on cDNA derived from peripheral blood mononuclear cells reveals nullification of miR-142-3p and miR-142-5p expression in miR-142$^{-/-}$ animals. Representative results from one of two independent experiments are shown (mean + SEM) with three animals in each group. \***p<0.0005. (**C**) Beta-galactosidase activity in *ex vivo* hematopoietic cell populations isolated from miR-142$^{+/+}$ (red), and miR-142$^{+/-}$ (blue) mice as determined by fluorescence-activated cell sorting (FACS) of Fluorescein Di-beta-D-Galactopyranoside-treated cells. Assayed cell types include T-cells (CD4$^+$ and CD8$^+$) and B-cells (immature, marginal and mature) derived from the spleen, granulocytes, monocytes (Gr1$^+$ and Gr1$^-$), natural killer (NK) cells, and megakaryocytes (MKs) derived from the BM. (**D** and **E**) Significant decrease in circulating red blood cells (RBC, panel **D**) and white blood cells (WBC, panel **E**) in 2-month-old miR-142$^{-/-}$ mice. Representative results from one of two independent experiments are shown (mean + SEM) with at least five animals in each group. \*p<0.05; \*\*p<0.005. (**F**) Significant decrease in circulating platelet numbers in 2- and 12-month-old miR-142$^{-/-}$ mice. Representative results from one of two independent experiments are shown (mean + SEM) with at least five animals in each group. \*\*p<0.005; \***p<0.0005. (**G**) Enlarged mean platelet volume (MPV) in 2- and 12-month-old miR-142$^{-/-}$ mice. Representative results from one of two independent experiments are shown (mean + SEM) with at least five animals in each group. \*\*p<0.005; \***p<0.0005.

The following source data and figure supplements are available for figure 1:

**Source data 1**. Mendelian distribution of miR-142 intercrosses.

**Figure supplement 1**. miR-142 hematopoietic-intrinsic expression is required for thrombopoiesis.

In contrast, the direct descendants of PreMegEs, namely the unipotent MK-progenitors (MkPs), which further give rise to mature MKs, were significantly increased in miR-142$^{-/-}$ animals, relative to WT controls (*Figure 2A,B*). Intriguingly, the expression levels of regulatory markers of MK development remained largely unchanged in miR-142$^{-/-}$ PreMegEs and MkPs, relative to WT controls (*Figure 2C,D*).

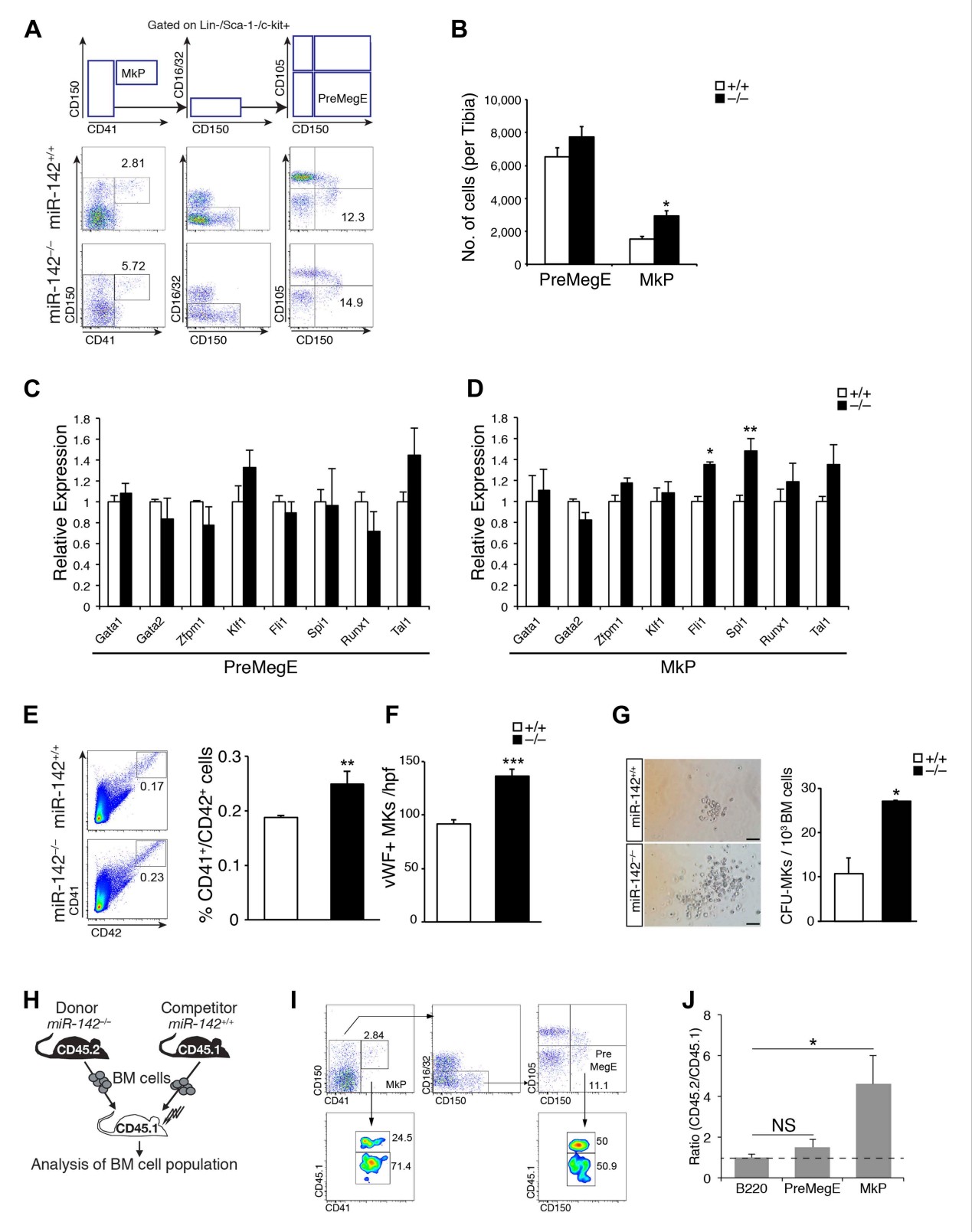

**Figure 2**. Perturbed myeloerythroid development in the absence of miR-142. (**A**) Diagram of gating strategy used to define the myeloerythroid progenitor populations (top panel), and representative FACS profiles of mutant miR-142−/− and WT BM cells (bottom panels). (**B**) Flow cytometric analysis of whole tibia BM-resident mega-erythroid progenitors (PreMegE; lin−c-kit+CD150+CD105−CD41−) and MK progenitors (MkP; lin−c-kit+CD41+), of *Figure 2. Continued on next page*

*Figure 2. Continued*

2-month-old animals. miR-142 deficiency results in increased MkP numbers and only modest, insignificant, changes in PreMegEs. Representative results from one of two independent experiments (mean + SEM) with at least three animals per group. *p<0. 05. (C and D) qPCR expression analysis of critical regulators of MK development: GATA binding protein 1 (Gata1), GATA binding protein 2 (Gata2), zinc finger protein, multitype 1 (Zfpm1), Kruppel-like factor 1 (Klf1), Friend leukemia integration 1 (Fli1), spleen focus forming virus proviral integration oncogene (Spi1), Runt-related transcription factor 1 (Runx1), and T cell acute lymphocytic leukemia 1 (Tal1) in miR-142$^{-/-}$ PreMegEs (C) and MkPs (D), relative to controls. Data normalized to Hprt expression and to the mRNA expression in WT controls and presented as mean + SEM. *p<0.05; **p<0.005. (E) Left panel, representative FACS profiles of mutant miR-142$^{-/-}$ and WT BM cells. Right panel, gating CD41$^+$/CD42$^+$ cells out of total BM, reveals increased mutant miR-142$^{-/-}$ MK numbers relative to WT controls. Representative results from one of two independent experiments (mean + SEM), at least three animals in each group. **p<0.005. (F) Increased numbers of von Willebrand factor (vWF)-positive MKs per high power field (hpf) in miR-142$^{-/-}$ BMs, relative to WT controls. Representative results from one of two independent experiments (mean + SEM), four cross-sections measured from each group. ***p<0.0005. (G) Left panel, CFU–MK assays demonstrate increased miR-142$^{-/-}$ MK numbers per colony. Scale bars, 50 μm. Right panel, increased numbers of MkPs in miR-142$^{-/-}$ BM, revealed by CFU–MK colony formation assay. Representative results from one of two independent experiments (mean + SEM), two biological samples in each group. *p<0. 05. (H) Schematic representation of the experimental design for competitive repopulation assay. (I) Representative FACS profiles for chimeric animals in competitive repopulation assay: [miR-142$^{-/-}$ (CD45.2)/WT (CD45.1) > WT (CD45.1)]. Flow cytometry performed 6 weeks after transplantation. (J) Quantification of CD45.2/CD45.1 ratios, calculated for each gate in three different animals. Dashed line indicates ratio of 1. Values >1 indicate that miR-142$^{-/-}$ (CD45.2) mutant cells out-compete WT (CD45.1) cells, whereas values <1 reveal the advantage of WT (CD45.1) cells. The CD45.2/CD45.1 ratio for B220-positive cells served as control. Representative results from one of two independent experiments (mean + SEM), three animals in each group. *p<0.05.

The following figure supplements are available for figure 2:

**Figure supplement 1**. Gene expression in miR-142$^{-/-}$ MKs.

**Figure supplement 2**. Splenomegaly and increased numbers of splenic MKs in miR-142$^{-/-}$ mice.

---

The observed expansion of MkPs in miR-142 mutants was further supported by a concordant elevation in the numbers of CD41$^+$/CD42$^+$ miR-142$^{-/-}$ MKs relative to WT controls (*Figure 2E*).

We next investigated the numbers of MKs *in situ,* by immunohistochemical staining of femoral BM sections against von Willebrand Factor (vWF), which is specifically expressed in MKs from an early stage of differentiation (*Tomer, 2004*). miR-142$^{-/-}$ mice displayed a ~50% increase in the numbers of vWF-positive MKs, relative to WT littermates (*Figure 2F*). Noteworthy, miR-142$^{-/-}$ mice exhibited splenomegaly and a marked elevation in the number of splenic MKs, relative to control littermates, suggesting extramedullary thrombopoiesis (*Figure 2—figure supplement 2*).

To further confirm the expansion of miR-142$^{-/-}$ MkPs, we employed a colony forming unit-megakaryocyte (CFU-MK) assay that quantifies the MkP numbers in the BM. We observed higher numbers of CFU-MK colonies of miR-142$^{-/-}$ BMs, relative to WT controls (*Figure 2G*, right panel). Furthermore, each miR-142$^{-/-}$ colony typically harbored more cells than control colonies (*Figure 2G*, left panel).

To elucidate whether MkP expansion represented a cell-intrinsic phenomenon, we employed a competitive repopulation experiment (*Figure 2H*). Thus, we injected CD45.2$^+$/miR-142$^{-/-}$ and congenic CD45.1$^+$/WT BM cells in equal numbers into lethally irradiated CD45.1$^+$ recipient mice. The reconstituted BM populations were analyzed for 6 weeks following transplantation. Cells expressing the pan-B-cell marker, B220, were equally represented by CD45.2$^+$/miR-142$^{-/-}$ and CD45.1$^+$/WT genotypes and served as engraftment controls. CD45.2$^+$/miR-142$^{-/-}$ PreMegE levels showed a mild increase relative to CD45.1$^+$/WT counterparts, confirming that this population is not appreciably affected by the loss of miR-142 (*Figure 2I,J*). In contrast, CD45.2$^+$/miR-142$^{-/-}$ MkPs were over-represented in chimeric BMs at a ratio of ~5:1 relative to CD45.1$^+$/WT MkPs (*Figure 2I,J*). Thus, the MkP expansion observed in miR-142$^{-/-}$ BM is cell-autonomous. Taken together, these data suggest that miR-142 activity regulates the differentiation of the MK lineage in a cell-intrinsic manner.

## Incomplete MK maturation in the absence of miR-142

The observed elevation in miR-142$^{-/-}$ MK frequency was unexpected, because increased MK numbers are usually correlated with higher platelet counts (*Schafer, 2004*). Thus, since the pronounced thrombocytopenia in miR-142-deficient mice was not caused due to a lack of MKs, we hypothesized that it may result from a block in MK maturation. An initial clue that miR-142$^{-/-}$ MKs were premature, came from the observation that the average size of vWF-positive miR-142$^{-/-}$ MKs in the femoral BM was

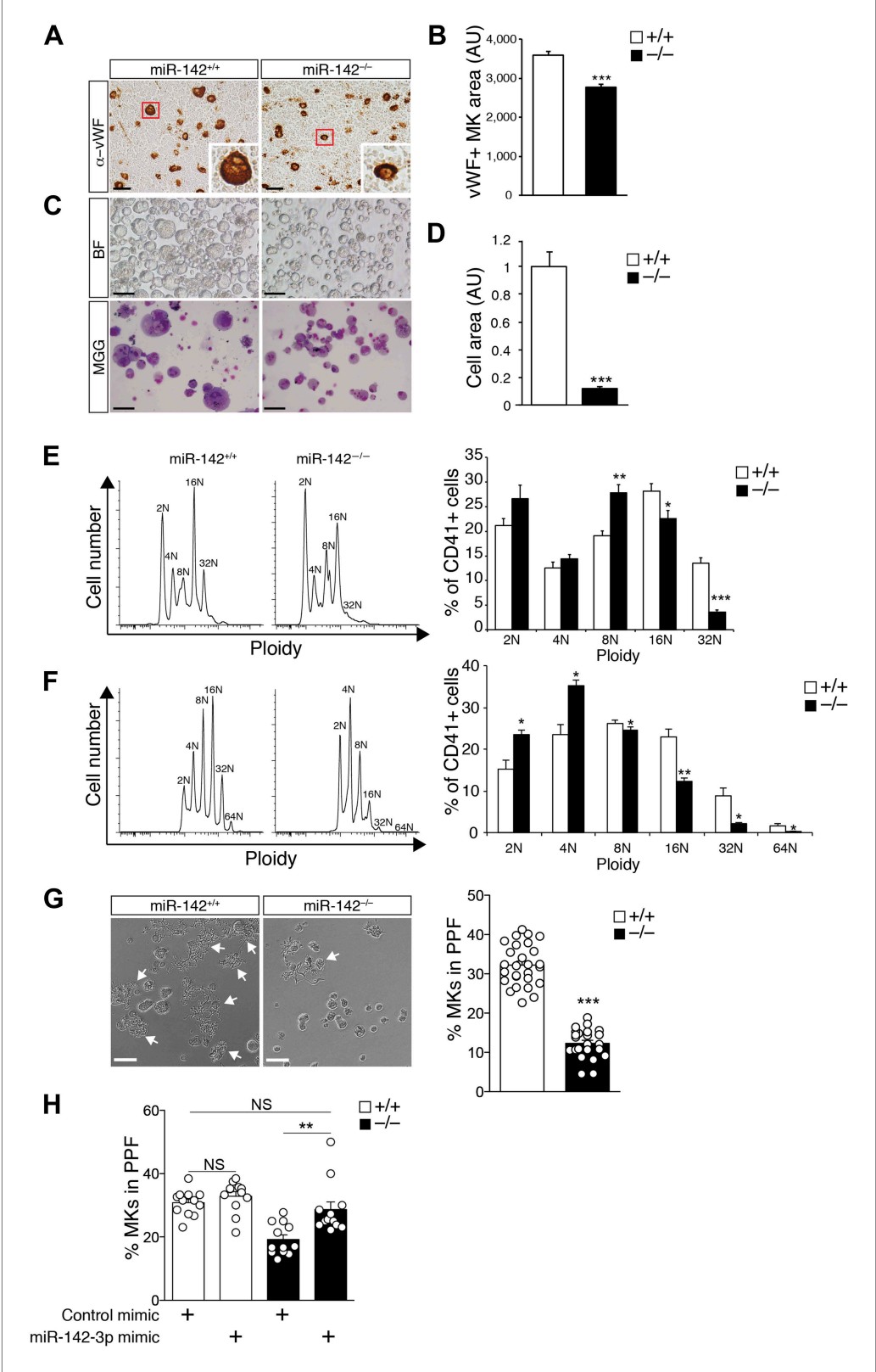

**Figure 3**. Impaired maturation of miR-142−/− MKs. (**A**) Representative BM sections of miR-142−/− vs WT controls, stained with anti-von Willebrand Factor (anti vWF). In the bottom right corner of each section there is an enlarged image of a representative vWF-positive MK. Scale bars, 50 μm. (**B**) Reduced diameter of vWF-positive MKs in
*Figure 3. Continued on next page*

*Figure 3. Continued*

miR-142⁻/⁻ BM relative to WT controls. Representative results from one of two independent experiments (mean + SEM). Data collected from four cross-sections measured and >100 cells per group. ***p<0.0005. (**C**) Representative brightfield (BF, top panel) and May–Grünwald Giemsa-stained (MGG, bottom panel) micrographs of FL-derived MK cultures, following enrichment by a BSA density gradient. Scale bars, 100 µm. (**D**) Size quantification of FL-derived MK, measured as pixel area and normalized to WT controls, reveals reduction of miR-142⁻/⁻ MK cell area. Representative results from one of two independent experiments (mean + SEM), >20 cells measured per group. ***p<0.0005. (**E**) Representative FACS plot of DNA content analysis for FL-derived MKs stained for CD41 and DAPI (left) and quantification of ploidy in FL-derived MKs, presented as a percentage of cells out of total CD41⁺ cells (right). AU, arbitrary units. Representative results from one of two independent experiments (mean + SEM) ≥4 animals in each group. *p<0.05; **p<0.005; ***p<0.0005. (**F**) Representative FACS plot of DNA content analysis for BM-derived MKs stained for CD41 and DAPI (left) and quantification of ploidy in BM-derived MKs, presented as percentage of cells out of total CD41⁺ cells (right). AU, arbitrary units. Representative results from one of two independent experiments are shown (mean + SEM) ≥4 animals in each group. *p<0. 05; **p<0.005. (**G**) Left panel: representative micrographs of proplatelet formation (PPF) in FL-derived WT or miR-142⁻/⁻ MKs (white arrows denote MKs extending proplatelets). Scale bars, 50 µm. Right panel: quantification revealed reduced PPF levels in miR-142⁻/⁻ FL-derived MKs. Representative results from one of two independent experiments a (mean ± SEM), three animals in each group, each animal represented by 7–9 experimental repeats in distinct wells and each dot is a representation of a single well. ***p<0.0005. (**H**) Re-introduction of miR-142-3p using dsRNA mimetics was sufficient to restore WT PPF levels to miR-142-deficient differentiated MKs. Overexpression of miR-142-3p mimic did not result in significant increase in PPF levels in WT MKs. Each dot represents data from a single well. **p<0.005; (NS) not significant.

The following figure supplements are available for figure 3:

**Figure supplement 1**. miR-142-3p expression in differentiated MKs following transfection of dsRNA mimetics. qPCR expression analyses confirm overexpression of miR-142-3p in WT and miR-142⁻/⁻ FL-derived MKs transfected with miR-142-3p mimic, relative to MKs transfected with control mimics.

smaller than that of WT MKs. Indeed, miR-142⁻/⁻ MKs showed a ~25% reduction in sectional area, relative to WT counterparts (*Figure 3A,B*).

We then differentiated MKs from E14.5 fetal liver (FL), under defined *ex vivo* conditions, as previously described (*Shivdasani and Schulze, 2005*). After 4 days in culture, FL-derived miR-142⁻/⁻ MKs exhibited smaller cell size, compared to control MKs (*Figure 3C,D*).

Polyploidization is an additional important feature of MK maturity, which is associated with effective platelet production (*Levine et al., 1982*; *Mattia et al., 2002*; *Ravid et al., 2002*; *Lee et al., 2009*). We therefore tested the number of MK nuclei by flow cytometry. CD41⁺ BM-derived miR-142⁻/⁻ MKs, exhibited reduced overall ploidy (*Figure 3E*). Furthermore, the fraction of mature (≥16N) miR-142⁻/⁻ MKs was significantly diminished, relative to WT controls, whereas the percentage of low ploidy immature MK forms (≤8N) was higher in miR-142⁻/⁻ BM (*Figure 3E*). Similar data were gained by *ex vivo* differentiation of FL-derived MKs, whereby high ploidy number (>32) was observed in only 4% of the miR-142⁻/⁻ MK, relative to 12% in control MKs (*Figure 3F*). Thus, miR-142 is essential for normal endomitosis and reduced miR-142⁻/⁻ platelet numbers might result from accumulation of immature, low-polyploid MKs that are poor producers of platelets (*Mattia et al., 2002*).

Proplatelet formation (PPF) represents the final phase of MK maturation, culminating in platelet release into the bloodstream (*Machlus and Italiano, 2013*). To analyze whether miR-142 is involved in this process, we performed an *ex vivo* PPF study on FL-derived MKs. Remarkably, we observed a striking threefold reduction in miR-142⁻/⁻ MKs that were extending proplatelets, relative to control MKs (*Figure 3G*). We next re-introduced miR-142-3p into differentiated MKs, using dsRNA mimetics (*Figure 3—figure supplement 1*). The introduction of miR-142-3p was sufficient to recapitulate WT PPF levels in cells that are genetically miR-142-deficient (*Figure 3H*). Conversely, overexpression of miR-142-3p mimics in WT MKs did not yield any significant increase in PPF levels. Thus, miR-142-3p activity is essential for proper MK maturation and its loss results in defective platelet biogenesis. Furthermore, since re-introduction of miR-142-3p into miR-142⁻/⁻ differentiated MKs restored functional identity, we conclude that there might be a continuous requirement for miR-142-3p activity to maintain MK maturity.

## Disturbed actin cytoskeletal dynamics in the absence of miR-142

Dynamic rearrangement and organization of MK cytoskeletal structures is essential for normal mega-karyopoiesis, endomitosis, and platelet production (*Hartwig and Italiano, 2006*; *Lordier et al., 2008*; *Thon and Italiano, 2012*). We therefore sought to characterize by confocal microscopy filamentous actin (F-actin) and tubulin in FL-derived MKs that were allowed to adhere to fibronectin for 3 hr. This analysis revealed a markedly disturbed cytoskeletal organization in miR-142$^{-/-}$ MKs compared to WT counterparts (*Figure 4A*) and mutant cells displayed a more immature circular profile than WT counterparts (*Figure 4B*).

Accordingly, ImagestreamX flow cytometry revealed a lower ratio of filamentous to globular (F/G) actin in miR-142$^{-/-}$ CD61$^+$ MKs than in control MKs (*Figure 4C,D*). This analysis reaffirmed that miR-142$^{-/-}$ MKs are smaller in size, more circular and exhibited a more homogenous F-actin distribution than controls (*Figure 4E–G*).

Finally, we performed a cytochalasin D (CytoD) washout study, which followed the re-assembly of actin filaments after forced depolymerization (*Figure 4H*). The number of miR-142$^{-/-}$ MKs that were able to create stress-fibers following treatment with CytoD was significantly lower, relative to control MKs (*Figure 4I*). Collectively, these data reveal that miR-142 is pivotal for normal actin dynamics and architecture in MKs.

## miR-142-3p targets a group of actin cytoskeleton regulators during megakaryopoeisis

To determine the molecular mechanism for miR-142-mediated control of megakaryocytic development, we performed a genome-wide study for the identification of differentially expressed genes in miR-142$^{-/-}$ vs WT differentiated MKs. This study revealed that roughly 800 mRNAs were significantly deregulated due to loss of miR-142 (*Figure 5A*). Sylamer analysis (*van Dongen et al., 2008*) of micro-array data, from WT and miR-142$^{-/-}$ FL-derived MKs, uncovered a highly significant enrichment for the 7- and 8-mer seeds of miR-142-3p among genes that were up-regulated in miR-142$^{-/-}$ MKs (Blue lines; *Figure 5B*). However, such enrichment was not evident for any other miRNA including the sister miRNA, miR-142-5p (Green lines; *Figure 5B*), further substantiating that miR-142-3p is the dominant functional miRNA from the pre-miR-142 hairpin. Accordingly, we depicted significant up-regulation of miR-142-3p targets within the set of TargetScan predicted targets (*Friedman et al., 2009*; *Figure 5—figure supplement 1*).

Gene ontology (GO) analysis using the database for annotation, visualization, and integrated discovery (DAVID) algorithm (*Huang da et al., 2009a*, *2009b*) revealed perturbations of a variety of cellular processes in miR-142$^{-/-}$ MKs (*Figure 5—source data 1*). To enhance the power of our analysis, we next focused on targets that were commonly up-regulated in miR-142$^{-/-}$ MKs as well as in miR-142$^{-/-}$ CD24$^+$ and CD24$^-$ in vitro-derived DCs that we previously characterized (*Mildner et al., 2013*). Forty genes were found to be commonly up-regulated in all three miR-142$^{-/-}$ hematopoietic cell types (*Figure 5C*, *Figure 5—source data 2*). This list is highly enriched with direct miR-142-3p putative targets (12 out of 40 genes). Subsequent DAVID analysis performed on these genes uncovered a significant enrichment for actin- and cytoskeleton-related functional GO categories (*Figure 5C*). Indeed, half of the dozen predicted miR-142-3p targets encode for pivotal actin cytoskeleton-associated proteins, including Cofilin-2 (Cfl2), Glucocorticoid receptor DNA binding factor 1 (Grlf1), Biorientation of chromosomes in cell division 1 (Bod1), Integrin alpha V (Itgav), Twinfilin-1 (Twf1), and Wiskott–Aldrich syndrome-like (Wasl). Intriguingly, the latter was recently suggested as a potential target of miR-142-3p in the process of actin-mediated mycobacterial infection (*Bettencourt et al., 2013*). The up-regulation of these six miR-142-3p targets in miR-142$^{-/-}$ MKs, was further confirmed by qPCR (*Figure 5D*). Among these targets, Cfl2 expression was also found to be up-regulated in miR-142$^{-/-}$ MkPs (*Figure 5—figure supplement 2*). Introducing of miR-142-3p into differentiated MKs, using dsRNA mimetics, was sufficient to normalize the expression levels of representative miR-142-3p targets, Wasl and Cfl2 in miR-142$^{-/-}$ MKs (*Figure 5E*). Western blot analysis revealed up-regulation of the protein products of Wasl and Cfl2 in miR-142$^{-/-}$ MKs, further substantiating them as bona fide miR-142-3p targets (*Figure 5F–G*). Using luciferase reporter assays, we showed that the mRNAs of Wasl, Cfl2, Grlf1, Itgav and Twf1 are directly targeted by miR-142-3p (*Figure 5H-J*, *Figure 5—figure supplements 3 and 4*). Furthermore, the quantitative contributions of individual binding sites within the target 3' UTR was revealed by stepwise loss of miR-142 regulation, when miR-142-3p binding sites were sequentially mutated (*Figure 5H–J*, *Figure 5—figure supplements 3 and 4*).

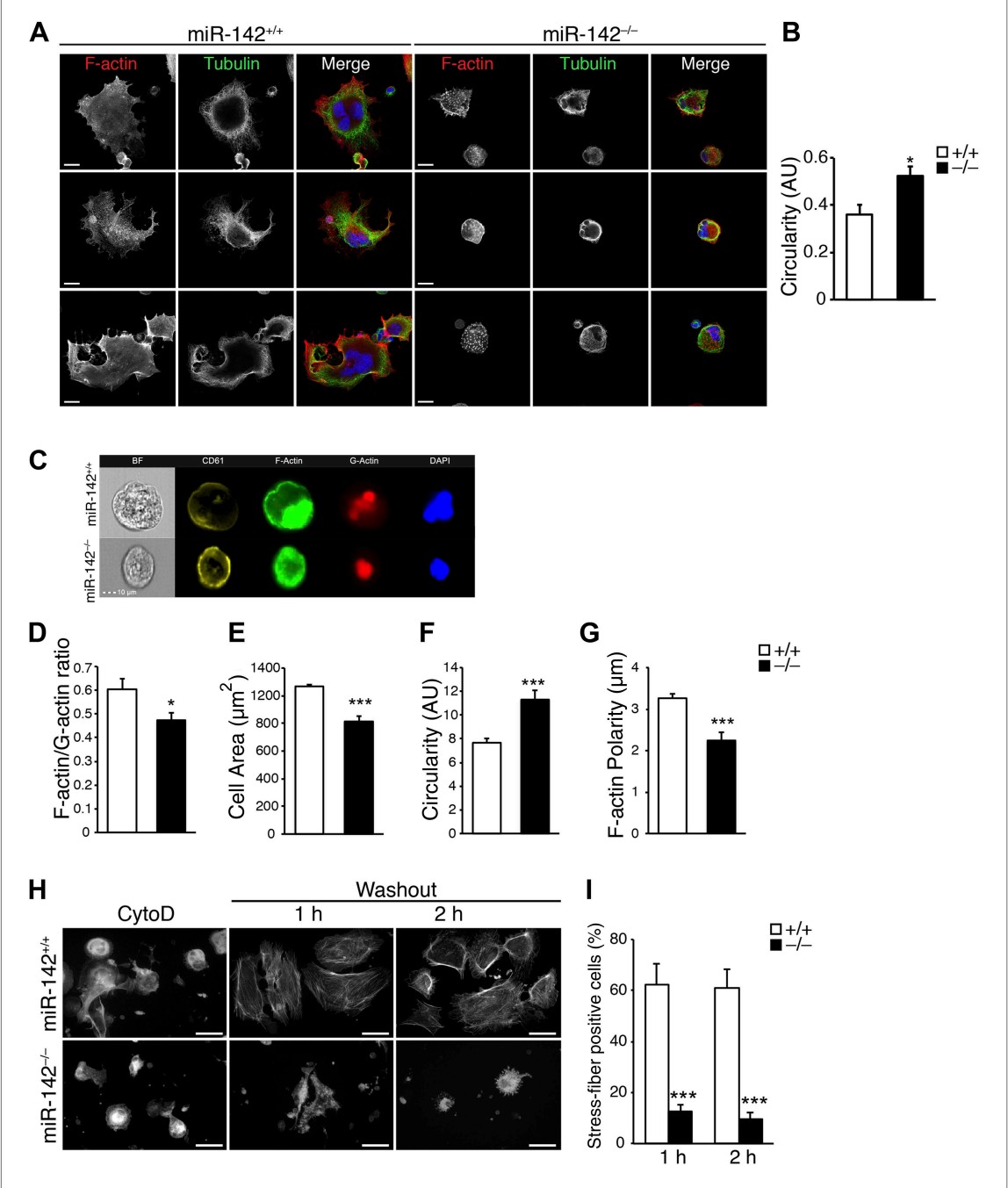

**Figure 4**. Disturbed actin cytoskeletal architecture and dynamics in the absence of miR-142. (**A**) Representative micrographs of WT and miR-142⁻/⁻ FL-derived MKs, cultured for 5 days with TPO and subsequently plated on fibronectin-coated cover-slips for 3 hr. F-actin (phalloidin, red), tubulin (green). The merged panels also depict DAPI in blue. Scale bars, 20 μm. (**B**) Circularity of FL-derived MKs (on an arbitrary scale of 0–1), measured using ImageJ software. miR-142⁻/⁻ MKs were more circular than WT controls, reflecting immaturity and relative deficiency of proplatelet-like structures. Representative results from one of two independent experiments (mean + SEM), >100 cells measured per group ***p<0.0005. (**C**) Representative flow cytometry-based single-cell images of FL-derived MKs as obtained by ImagestreamX flow cytometer, stained with anti - CD61 antibody (yellow), FITC-Phalloidin (F-actin, green), Alexa594-DNaseI (G-actin, red) and DAPI (blue). Scale bar, 10 μm. (**D–G**) Reduced F-actin/G-actin ratio in FL-derived MKs (**D**), reduced cell area (**E**), increased circularity (**F**), and increased F-actin polarity (**G**), in miR-142⁻/⁻ MKs relative to WT controls revealed by Imagestream analysis. Four animals per group (mean + SEM). *p<0.05; ***p<0.0005. (**H**) Representative micrographs of WT and miR-142⁻/⁻ MKs stained with Phalloidin–Rhodamine for detection of actin stress fibers after cytochalasin D (CytoD) washout. Left panel depicts MKs stained following 30 min of CytoD treatment. Middle and right panels depict MKs stained 1 hr and 2 hr after CytoD washout, respectively. Scale bars, 50 μm. (**I**) The fraction of WT MKs exhibiting stress fibers at 1 hr and 2 hr after CytoD washout is larger relative to miR-142⁻/⁻ MKs. Representative results from one of two independent experiments (mean + SEM), >50 cells counted in each group. ***p<0.0005.

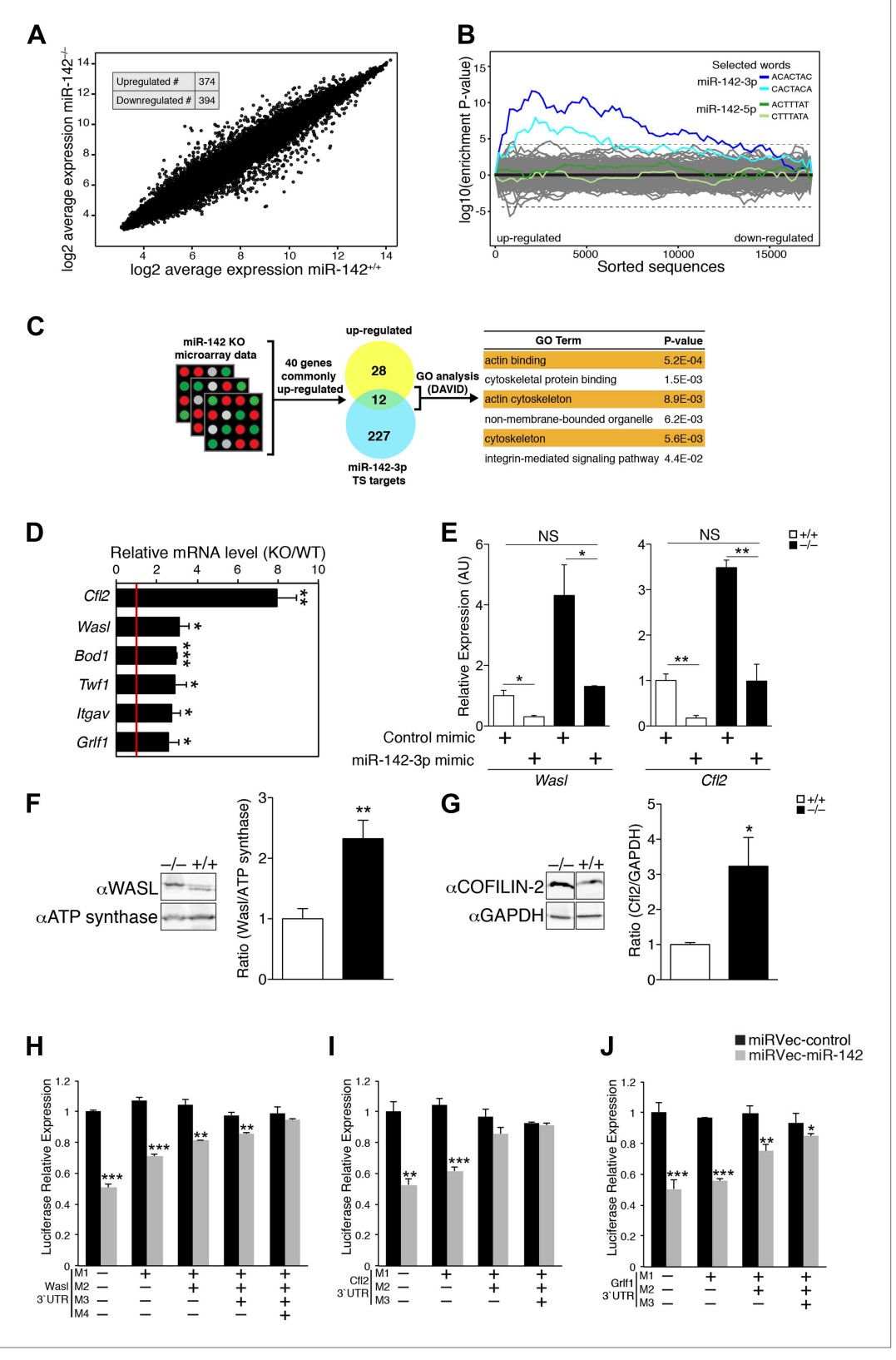

**Figure 5**. miR-142 Regulates a group of cytoskeletal regulatory genes during megakaryopoiesis. (**A**) A log2-scale scatter plot, presenting the expression of mRNAs from FL-derived WT MKs (x axis), and miR-142−/− MKs (y axis). The
*Figure 5. Continued on next page*

*Figure 5. Continued*

inset table depicts the number of genes that are significantly up- or down-regulated (> 2 fold change, p<0.05). (**B**) Enrichment landscape plot for all 876 7mer motifs complementary to canonical mouse miRNA seed regions, gained by Sylamer analysis (***van Dongen et al., 2008***). Sorted 17,000 gene list ordered from mostly up-regulated to mostly down-regulated in the miR-142$^{-/-}$ MKs on the x axis reveal the enrichment of only two motifs, which are both corresponding to the expected impact of miR-142-3p 'seed' on MK transcriptome (blue, 7mer–m8; light blue, 7mer–A1). miR-142-5p seed motifs are not enriched (green, 7mer-m8; light green, 7mer-A1). Horizontal dotted lines represent a Bonferroni-corrected p value threshold of 0.05. (**C**) A schematic representation of bioinformatic pipeline. Genes that were commonly and significantly up-regulated in three expression arrays of miR-142$^{-/-}$ hematopoietic cells (CD24$^+$ *in vitro*-derived dendritic cells [DCs], CD24$^-$ DCs and MKs), were superimposed with miR-142-3p TargetScan (TS)-predicted target genes. The resultant list was subjected to GO analysis using the DAVID bioinformatic tool. The majority of enriched GO categories were annotated to cytoskeletal and actin-binding genes. (**D**) qPCR expression analysis of a novel set of miR-142-3p target mRNAs: Cofilin-2 (Cfl2), Wiskott–Aldrich syndrome-like (Wasl), Biorientation of chromosomes in cell division 1 (Bod1), Twinfilin-1 (Twf1), Integrin alpha V (Itgav) and Glucocorticoid receptor DNA binding factor 1 (Grlf1) genes in miR-142$^{-/-}$ MKs, relative to controls, normalized to Hprt expression and to the mRNA expression in WT controls. Data are presented as mean + SEM. *p<0.05; **p<0.005; ***p<0.0005. (**E**) Reintroducing miR-142-3p using dsRNA mimetics was sufficient to restore Wasl (left panel) and Cfl2 (right panel) expression levels in miR-142-deficient differentiated MKs (black bars). In addition, overexpression of the miR-142-3p mimic resulted in significant reduction of Wasl (left panel) and Cfl2 (right panel) expression levels in WT MKs (white bars). *p<0.05; **p<0.005; (NS) not significant. (**F** and **G**) Western blot analysis of representative miR-142-3p target genes. Cell lysates from WT and miR-142$^{-/-}$ FL-derived MKs were subjected to SDS-polyacrylamide gel electrophoresis. WASL (**F**) and COFILIN-2 (**G**) were immunodetected and assessed by densitometry (right panel in **F** and **G**). ATP-synthase and GAPDH are indicators of protein loading levels, respectively (left panels in **F** and **G**). Representative results from one of two independent experiments (mean + SEM), four biological samples in each group. *p<0.05; **p<0.005. (**H–J**) Relative luciferase activity of reporters that harbor the 3'UTR of novel miRNA targets: Wasl (**H**), Cfl2 (**I**), and Grlf1 (**J**). Luciferase reporter activity is repressed by transfection of miR-142 expression vector (gray bars) in HEK-293T cells, whereas reporters that harbor a mutated version of the 3'UTR are insensitive to miR-142. Data normalized to the activity of firefly luciferase that is co-expressed from the dual reporter and to a negative control miRNA vector and presented as mean + SEM. *p<0.05; **p<0.005; ***p<0.0005.

The following source data and figure supplements are available for figure 5:

**Source data 1**. GO analysis for differentially regulated genes (>twofold) in miR-142$^{-/-}$ MKs.

**Source data 2**. Genes commonly up-regulated (>1.5-fold) in miR-142$^{-/-}$ MKs and DCs (CD24+ and CD24$^-$).

**Figure supplement 1**. Expression distribution plot of miR-142 putative targets.

**Figure supplement 2**. qPCR expression analysis of miR-142 putative targets in precursor cell populations.

**Figure supplement 3**. miR-142-3p directly regulates cytoskeletal genes.

**Figure supplement 4**. miR-142-3p directly regulates cytoskeletal genes.

Taken together, we discovered that the expression of a compound set of actin cytoskeleton regulators is post-transcriptionally controlled by miR-142-3p.

## miR-142-3p targets a battery of actin cytoskeleton regulators to facilitate proplatelet formation

miR-142-deficient MKs displayed perturbed actin filament dynamics and diminished proplatelet formation. This is presumably due to de-repression of several actin cytoskeleton components, including Wasl, Cfl2, Twf1, Itgav or Grlf1, which are all direct miR-142-3p targets. Thus, we hypothesized that knocking down these miR-142-3p targets may relieve the PPF defect in miR-142$^{-/-}$ MKs. For this purpose, we transduced FL-derived MKs with short hairpin RNA (shRNA)-expressing lentiviral vectors that effectively knocked-down targets with more than three miR-142-3p binding sites, namely Wasl, Cfl2 or Grlf1 (shWasl, shCfl2 and shGrlf1, respectively, *Figure 6—figure supplement 1*). A lentivirus encoding a shRNA directed against RFP (shRFP) was employed as a control. Following transduction,

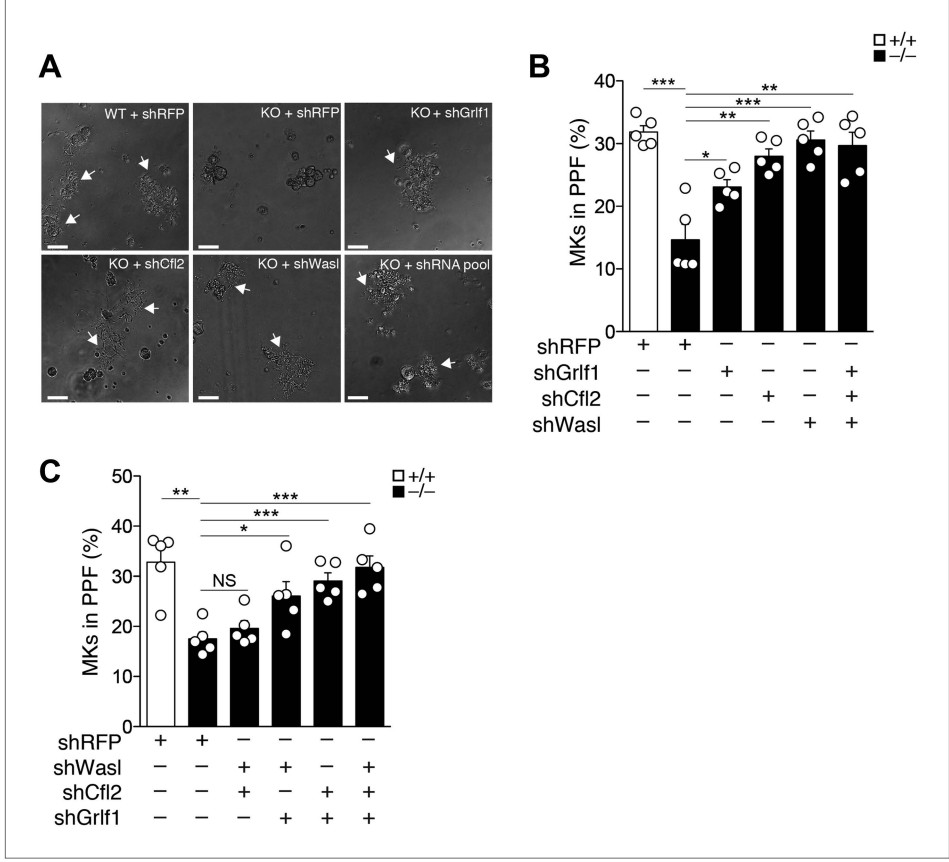

**Figure 6**. miR-142-3p targets a battery of actin cytoskeleton regulators to facilitate proplatelet formation. (**A**) Representative micrographs of WT and miR-142$^{-/-}$ (KO) FL-derived MKs, transduced with the indicated shRNA vectors and cultured for 48 hr with TPO. White arrows denote MKs extending proplatelets. Scale bars, 50 μm. (**B** and **C**) miR-142$^{-/-}$ MKs transduced with shRNAs targeting individual (**B**), paired (**C**), or a combined set (**B** and **C**) of miR-142-3p targets restore PPF levels. WT MKs were transduced with a control shRNA-targeting RFP. Representative results from one of two independent experiments (mean + SEM), five experimental repeats in each group (white dots). *p<0.05; **p<0.005; ***p<0.0005; (NS) not significant.

The following figure supplements are available for figure 6:

**Figure supplement 1**. Knockdown validation of miR-142-3p targets.

**Figure supplement 2**. Knockdown of miR-142-3p targets in WT MKs has no effect on proplatelet formation.

differentiation of MKs was induced and PPF was quantified 3 days later. The PPF defect was clearly evident in miR-142$^{-/-}$ MKs transduced with shRFP (**Figure 6A,B**). In contrast, miR-142$^{-/-}$ MKs that were transduced with shRNA directed against Cfl2, Wasl, Grlf1, displayed a prominent elevation of PPF levels, and approximated levels observed in control WT MKs transduced with shRFP (**Figure 6A,B**). This strong elevation in PPF levels was limited to miR-142$^{-/-}$ MKs, whereas in WT MKs, knockdown of miR-142-3p targets did not significantly alter PPF levels, except for Wasl-targeting shRNA that exhibited a modest increase in PPF efficacy following transduction (**Figure 6—figure supplement 2A**). Additional pairwise target comparison revealed compound relationships between miR-142-3p targets (**Figure 6C**; **Figure 6—figure supplement 2B**). Remarkably, knockdown of both Cfl2 and Wasl, which carry out opposing functions within the actin regulatory network, did not yield any significant increase in PPF levels in miR-142-deficient MKs (**Figure 6C**). Conversely, concomitant Cfl2 and Grlf1 knockdown, two proteins that are important for destabilizing actin polymers, enhanced PPF in miR-142 null MKs (**Figure 6C**). Lastly, a pool of shRNA against miR-142-3p targets was able to fully restore PPF capacity to WT levels (**Figure 6A–C**). These data demonstrate that the expression of a set of actin

cytoskeleton regulators should be tightly orchestrated by miR-142-3p in order to effectively promote platelet biogenesis.

## Discussion

Hematopoietic-specific miR-142 has emerged in recent years as a critical regulator of various blood and lymphoid cell lineages (*Gauwerky et al., 1989*; *Chen et al., 2004*; *Huang et al., 2009*; *Belz, 2013*; *Lagrange et al., 2013*; *Lu et al., 2013*; *Mildner et al., 2013*; *Nimmo et al., 2013*; *Sonda et al., 2013*; *Sun et al., 2013*; *Zhou et al., 2013*). Our analysis unveils the critical functions of miR-142 in the MK lineage. Using a recently-established mouse model, we show that deletion of the miR-142 allele results in pronounced thrombocytopenia. Our *in vivo* studies and culture assays reveal that proper miR-142 function is essential, in a cell-intrinsic manner, for MK maturation, including control of cell size, polyploidization and proplatelet elaboration.

Furthermore, MKs require sustained miR-142-3p expression, as re-introduction of synthetic miR-142-3p mimetics, even onto differentiated MKs, was sufficient to restore functional maturity.

Mechanistically, we demonstrate that miR-142-3p mediates the repression of an interconnected set of actin cytoskeleton regulators. These collectively contribute to MK maturation and their dysregulation is responsible for incomplete maturation observed in miR-142-deficient MKs. Very often, the sets of target genes that are predicted to be regulated by a particular miRNAs do not coalesce into coherent networks with distinct biological functions (*Stark et al., 2005*). However, our work provides a clear example of an individual miRNA that co-regulates a network of functionally-associated targets. Indeed, the ability of a miRNA to modulate the expression of multiple targets within the same pathway simultaneously was previously suggested in other cellular contexts. For instance, the bicistronic miR-143/145 miRNA gene cluster dictates smooth-muscle cell phenotypic switching by orchestrating the expression of a cadre of cytoskeletal remodeling regulators (*Xin et al., 2009*). Furthermore, miR-125 family members control hematopoietic stem cell pool size by targeting a cohort of proapoptotic genes (*Guo et al., 2010*; *Ooi et al., 2010*). Therefore, a single miRNA that cumulatively targets several nodes within the same biological circuit may serve as an effective means to control cellular behavior.

The actin cytoskeleton participates in a wide array of cellular functions, and the dynamic turnover between its F-actin and monomeric G-actin forms is regulated by a large number of actin-binding proteins. Within this cytoskeletal regulatory network, the set of genes targeted by miR-142-3p contains components with divergent functions. For example, Cfl2 and Twf1 participate in disassembly of actin filaments (*Bamburg, 1999*; *Moseley et al., 2006*). Likewise, Grlf1 is a GTPase activating protein that has been implicated in disruption of the organized actin cytoskeleton (*McGlade et al., 1993*). Wasl, on the other hand, promotes actin polymerization by catalyzing filament branching together with the Arp2/3 complex (*Rotty et al., 2013*). Thus, miR-142 deficiency destabilizes feedback loops required for actin filament homeostasis, stress fiber formation, and actin remodeling. This in turn impairs proplatelet generation and plausibly other MK-intrinsic cellular functions, such as endomitosis. It is also likely that additional miR-142-3p targets participate in the regulation of MK differentiation. For example, the over-representation of miR-142-deficient MkPs might result from a dysregulation of a distinct set of targets that are not necessarily related to actin regulation.

Repression of miR-142-3p actin-associated targets was found to be sufficient for restoring PPF levels in miR-142-deficient MKs. Interestingly, knocking down the same targets in WT MKs did not result in significant increase in PPF capacity over WT baseline levels. This might be related to a certain threshold in PPF levels that cannot be crossed, even when regulation of miR-142-3p is mimicked.

In summary, our analysis suggests a cardinal role for miR-142 in maturation of MKs, and in particular in controlling a network of chief actin regulators to facilitate MK terminal differentiation. The data challenges the prevailing paradigm, that miRNAs exert only subtle effects, often elicited by specific stressors (*Ebert and Sharp, 2012*; *Mendell and Olson, 2012*; *Pelaez and Carthew, 2012*), by providing *in vivo* evidence that genetic manipulation of a single miRNA may have a significant impact on cellular commitment and differentiation.

Several platelet disorders have been associated with mutations in genes involved in actin organization, including Wiskott-Aldrich syndrome protein (WASP) (*Massaad et al., 2013*) and Actinin alpha 1 (ACTN1) (*Kunishima et al., 2013*). Because miR-142 locus is involved in B-cell leukemogenesis (*Gauwerky et al., 1989*) and since miR-142 is necessary for CD4+ DCs development (*Mildner et al., 2013*) and MK differentiation, miR-142 may function as a broad hematopoietic pro-differentiation

factor. Thus, changes in miR-142 levels or activity may lead to platelet disorders or to hematopoietic malignancies.

## Materials and methods

### Generation of miR-142$^{-/-}$ mice

Mice strains were housed and handled in accordance with protocols approved by the Institutional Animal Care and Use Committee of WIS. To generate miR-142$^{-/-}$ mice, a gene-trapped embryonic stem cell clone (ES, C57BL/6J strain) from TIGM (College Station, TX), was chosen based on an insertion upstream of miR-142 hairpin. ES cells were microinjected into C57BL/6J host blastocysts. Chimeras and further transmission of the targeted allele through the male germline to heterozygous pedigree was confirmed by PCR analysis of genomic tail DNA. Homozygous and WT littermate mice were generated by additional intercrosses. For transplantation experiments, recipient mice were lethally irradiated, using a 10.5Gy cesium source. After 10 days of ciprofloxacin prophylaxis, approximately $5 \times 10^6$ cells were injected into the tail vein for repopulation of the hematopoietic system.

### Murine peripheral blood counts

About 100 μl whole blood was retro-orbitally drawn from age- and sex-matched miR-142$^{-/-}$ and WT littermates into glass capillary tubes that were pre-treated with 5 μl of 0.5M EDTA, to prevent coagulation. Complete blood count was performed on ADVIA 120 Hematology System (Siemens Healthcare, Erlangen, Germany) by American Medical Laboratories (Herzliya, Israel).

### Histology

Femora and spleens of age- and sex-matched miR-142$^{-/-}$ and WT littermates were excised after euthanasia and fixed overnight in 4% paraformaldehyde. Femora were then decalcified in 14% EDTA for 2–5 days. Specimens were dehydrated in graded ethanols, washed and processed into paraffin blocks. Longitudinal paraffin sections were stained with hematoxylin and eosin (H&E), May–Grünwald Giemsa or immunostained with anti-von Willebrand Factor (Dako, Agilent technologies, Santa Clara, CA). An Axioplan light microscope (Carl Zeiss, Oberkochen, Germany), equipped with an eyepiece graticule (grid), was utilized for quantification of MK numbers, per 20X magnification field, in 5 μm BM or spleen sections. MKs size was measured as the maximal diameter in 50 consecutive MKs in the BM of the distal femur, using the cellSens digital imaging software on an Olympus BX51 microscope at 40X magnification.

### Colony-forming unit assays

Mouse BM cells were harvested via flushing of the long bones with Dulbecco modified Eagle medium (DMEM; Invitrogen, Life Technologies, Carlsbad, CA) supplemented with 10% fetal bovine serum (FBS; Invitrogen, Life Technologies, Carlsbad, CA), was followed by filtering through a 70-μm nylon mesh cell strainer, to remove bone debris. BM mononuclear cells were cultured in MethoCult M3231 medium supplemented with 50 ng/ml Thrombopoietin (TPO; PeproTech, Rocky Hill, NJ), for 7 days according to the manufacturer's protocols (StemCell Technologies, Vancouver, British Columbia, Canada). Colonies containing >3 MKs were counted as CFU–MKs. Duplicate assays were performed for each mouse. At least two mice were analyzed for each sample group.

### Primary megakaryocyte microscopy and proplatelet formation

Mouse FLs, collected on E14.5, were processed into single-cell suspension by successively passing through 18- 21- and 23-gauge needles. Cells were then cultured in DMEM supplemented with 10% FBS, 50 ng/ml murine TPO, 2 mM L-glutamine and penicillin–streptomycin. After 4–5 days, MKs were purified using a discontinuous gradient of bovine serum albumin (BSA, 3%, 1.5%, and 0%, Sigma-Aldrich, St. Louis, MO). About $1–2 \times 10^4$ purified MKs were cultured per 6.4 mm diameter well in suspension in flat bottom 96-well plates. After 16 hr of incubation, the fraction of proplatelet-forming MKs per well was scored with a light microscope under label-blinded experimental designs.

For Immunocytofluorescence, FL-derived MKs were allowed to adhere to fibronectin-coated cover slips for 3 hr. Cover slips were rinsed with PBS, fixed with 3.7% formaldehyde, and permeabilized with 0.1% Triton X–100 (Sigma-Aldrich, St. Louis, MO). Cells were blocked with PBS, 2% BSA and incubated with Phalloidin–Rhodamine (gift from Benny Geiger, Weizmann Institute of Science) and anti-alpha–Tubulin

antibody (gift from Alexander Bershadsky, Weizmann Institute of Science) for 1 hr. Following blocking, 4'6-Diamidino-2-phenylindole dihydrochloride (DAPI) was added for 5 min before slides were mounted with Immu-Mount (Thermo Scientific). For actin dynamics experiments, MKs were treated after plating for 30 min with 1 mM cytochalasin D (gift from Benny Shilo, Weizmann Institute of Science) and subsequently the drug was washed-out by three medium changes. Cells were fixed with 4% paraformaldehyde at 1 hr or 2 hr after washout and then permeabilized with 0.2% Triton X-100 in PBS. Stress fibers were stained with FITC-conjugated Phalloidin (Sigma-Aldrich, St. Louis, MO) and fluorescent micrographs were captured with a Zeiss LSM510 Laser Scanning confocal microscope.

## Flow cytometry

Flow cytometric analysis was performed on a LSRII flow cytometer (BD Biosciences, San Jose, CA) with FlowJo Version 8.8.7 software (TreeStar, Ashland, OR). BM from 6- to 8-weeks-old mouse femora and tibiae or FL-derived MKs were treated with red blood cell lysis buffer (Ammonium-Chloride-Kalium, ACK, 0.15 M NH4Cl, 0.1 M KHCO3, 1 mM EDTA in PBS). Cells were stained with PE-conjugated anti-CD41 antibody (Abcam, Cambridge, England), FITC-conjugated anti-CD42b (Emfret), or APC-conjugated anti-CD61 antibody. For DNA content analysis, cells were further fixed with 2% paraformaldehyde, stained with 1 μg/ml DAPI, 0.1% BSA (Sigma-Aldrich, St. Louis, MO) and 0.1% Saponin (Sigma-Aldrich, St. Louis, MO). For detailed analysis, freshly obtained BM cells were stained with APC anti-CD150, PE anti-CD41 (Abcam, Cambridge, England), FITC anti-Sca-1, Brilliant Violet anti-CD117, PE-Cy7 anti-CD105, Alexa700 anti-CD16/32, biotin-labeled lineage cell detection cocktail (CD4, CD8α, B220, Ter119, Gr1, CD11b), and streptavidin PerCP–Cy5.5. All antibodies were from BioLegend or eBioscience, unless otherwise indicated. MK-erythroid bipotent progenitors, PreMegE, were gated by lin⁻c-kit⁺CD150⁺CD105⁻CD41⁻ or lin⁻sca1⁻c-kit⁺CD16/32⁻, and MK progenitors, MkP, were gated by lin⁻sca1⁻c-kit⁺CD41⁺ (*Pronk et al., 2007*; *Pronk and Bryder, 2011*). Sorted PreMegE and MkP were collected and RNA was extracted using RNeasy micro kit (Qiagen, Venlo, Netherlands).

Quantification of actin intensity and morphocytometry was performed with ImagestreamX flow cytometer and IDEAS 6.0 software (Amnis Corp., Seattle, WA). 2 × 10⁴ FL-derived MKs were stained with APC-conjugated CD61 (Biolegend, San Diego, CA), fixed using the Cytofix/Cytoperm kit (BD Biosciences, San Jose, CA), and further stained with FITC-conjugated-Phalloidin, Alexa594-conjugated DNaseI (Invitrogen, Life Technologies, Carlsbad, CA) and DAPI. Images were compensated for fluorescent dye overlap by using single-stain controls. Analysis was done on in-focus single cell images as previously described (*George et al., 2006*) with single cell gating, using the area and aspect ratio features. Cell area was calculated in square microns from brightfield images. Circularity was calculated as average distance of object boundary from center, divided by the variation of this distance. Thus, shapes approximating circle exhibited low variation and gained higher values (in arbitrary units). Actin polarity was calculated using the Delta Centroid XY feature, which calculates the distance (in microns) between image center (brightfield) and the intensity-weighted actin image center (higher values indicate increased polarity). The F/G actin ratio was calculated by dividing the corresponding pixel intensities for each cell.

## Lentiviral vector production and cell transduction

HEK-293T cells were transfected by calcium phosphate with pLKO.1 encoding shRNAs for knockdown of Wasl, Grlf1, Cfl2, or RFP (TRC, Broad Institute of MIT and Harvard, Cambridge, MA), and lentivirus packaging plasmids (pPAX2 pMD2). Lentivirus supernatants were stored, and for knockdown efficacy assessment, lentiviral particles were added at multiplicity of infection (MOI) of 2, onto 5 × 10⁴ NIH-3T3 cells in 16-mm wells that were incubated with medium containing 8 μg/ml polybrene (Sigma-Aldrich, St. Louis, MO). Selection of transduced cells was performed with puromycin (2 μg/ml) that was added from the second day and until cells were harvested on day 5.

For MK transduction, approximately 2 × 10⁵ FL cells were cultured up to 4 days in medium supplemented with 50 ng/ml TPO. On the fourth day, cells were purified using a BSA gradient and 1 × 10⁴ cells were placed in 6.4-mm rounded wells in suspension with medium containing 8 μg/ml polybrene that was freshly supplemented with TPO. Lentiviral particles were added at MOI of 25 and transducted through centrifugation at 900×*g*, 32°C for 90 min. For mock transduction, an equivalent volume of medium was added. Cells were incubated at 37°C overnight, washed in PBS and split into five different 6.4-mm wells in a 96-well flat bottom suspension plate.

## Preparation of b-PEI₂₅-CAN-γ-Fe₂O₃ nanoparticles (NPs) and miRNA oligonucleotides transfection

Ultra-small core-shell maghemite nanoparticles consisting of a cerium [Ce(III/IV)] cation-doped CAN-$\gamma$-Fe$_2$O$_3$ core and a coordinated branched polyethylenimine (b-PEI$_{25}$) shell (25 kDa) have been prepared according to Israel et al. (patent application PCT/IL2014/050064). Typically, a CAN-$\gamma$-Fe$_2$O$_3$ NPs aqueous suspension (1.0 ml, Fe: 1.93 mg/ml–1.93 mg total Fe, 0.0346 mmol Fe, ICP-AES measurement) was diluted in 25.0 ml double distilled water (ddH2O). For the nanoparticle functional shell, we used b-PEI$_{25}$, which enables electrostatic binding of nucleosides and endosome destabilization by osmotic imbalance, leading to subsequent release of RNA into cell cytoplasm. Therefore, 10.13 mg of polycationic branched b-PEI$_{25}$, (10.0 mg/ml stock solution, 0.4053 µmol, Sigma-Aldrich, St. Louis, MO) were added to CAN-$\gamma$-Fe$_2$O$_3$ NPs at a 1:5.25 ratio (wt/wt). Mild b-PEI$_{25}$ coordinated coating was accomplished by overnight orbital shaking at room temperature. The resulting crude core-shell b-PEI$_{25}$-CAN-$\gamma$-Fe$_2$O$_3$ nanoparticles were washed three times in 10 ml ddH$_2$O using an Ultra-15 Amicon centrifugal filter (100K, EMD-Millipore, Billerica, MA) operated for 5 min at 4,000 rpm. Then, a size exclusion process was performed by centrifugation (8,000 rpm, 16 min, 18°C and 7,000 rpm, 10 min, 18°C) afforded the corresponding cleaned b-PEI$_{25}$-CAN-$\gamma$-Fe$_2$O$_3$ nanoparticles. Selected nanocomposite characterization of such functional nanoparticles disclosed respective average TEM/DLS NP diameters of 6.86 ± 1.55 and 82.90 nm ± 1.26 (DLS, PDI: 0.195). NP ξ potential (ddH2O) is +31.1 mV. TGA weight loss (N$_2$ atmosphere, 200–410°C temperature range) is 73.62%.

For MK transfection, 0.49 µg (100 nM) of miR-142-3p mimics dsRNA oligonucleotides, or control sequence (Integrated DNA Technologies, Inc., Coralville, IA) were mixed with b-PEI$_{25}$-CAN-$\gamma$-Fe$_2$O$_3$ nanoparticles at a 0.315 Fe/dsRNA wt/wt ratio, incubated 15 min at RT, and then, transfected to 3 × 10$^4$ FL-derived MKs in 35-mm plates. mmu-miR-142-3p guide sequence is: U*G*UAGUGUUUCCUACUUUAUmGmGA. An extensively-modified passenger strand sequence is: 5'-C3(spacer)/UmCCmAUmAAmAGmUAmGGmAAmACmACmUAmCA/3'-Cy5.5 (dye). 'm' indicates a 2'O-Methyl RNA and '*' indicates a phosphorothioate internucleotide linkage. Cells were then incubated at 37°C overnight, washed in PBS and further cultured in flat bottom 6.4-mm wells in the presence of TPO, for 48 hr after which PPF levels were scored, and RNA was extracted with RNeasy micro kit (Qiagen, Venlo, Netherlands).

## RNA analysis

Total RNA was isolated with Tri-Reagent (MRC) following manufacturer's instructions. RNA quality was assessed with ND–1000 Nanodrop (Peqlab) and on a 1.5% agarose gel prior to gene-expression profiling using the Mouse Genome Gene 1.0 ST Affymetrix Gene Chip according to the manufacturer's instructions. For real-time Quantitative (q) PCR, cDNA synthesis was carried out by using oligo d(T) primer (Promega) and SuperScript II reverse transcriptase (Invitrogen, Life Technologies, Carlsbad, CA), following manufacturer's instructions. qPCR analysis of mRNA expression was performed on a LightCycler 480 Real-Time PCR System (F. Hoffmann-La Roche Ltd, Basel, Switzerland), using KAPA SYBR FAST qPCR Kit (Kapa Biosystems, Wilmington, MA). Efficiency of each primer pair was confirmed by serial dilutions of templates. For quantification of mature mmu–miR–142 forms, cDNA synthesis was carried out by the miScript Reverse Transcription Kit and qPCR reaction utilized miScript SYBR Green PCR Kit (with miScript Universal Primer, Qiagen, Venlo, Netherlands). U6 and hypoxanthine phosphoribosyltransferase 1 (Hprt) were used as a reference for normalization of miRNA and mRNA levels, respectively. All primer sequences are provided in *Supplementary file 1*.

## Western blotting

For protein quantification, FL-derived MKs were lysed in radioimmunoprecipitation (RIPA) buffer with protease and phosphatase inhibitors. Protein concentration was determined using protein assay (Bio-Rad Laboratories Inc. Hercules, CA). 20 µg of proteins were separated by SDS polyacrylamide gel electrophoresis, electrotransferred onto 0.2-mm nitrocellulose membrane, blocked in TBS, 0.1% Tween20 and 5% dry milk for 1 hr and incubated overnight with primary Antibodies: anti-N-WASP/Wasl (4848s; 1:1000; Cell Signaling Technology), anti-Cofilin2 (ab96678; 1:1000; Abcam, Cambridge, England), anti-GAPDH (AM4300; 1:10,000; Ambion) and anti-ATP-synthase (MS507; 1:2000; MitoSciences, Eugene, OR). HRP-conjugated secondary antibody (Jackson ImmunoResearch Laboratories, West Grove, PA) was diluted in TBS, 0.1% Tween20. Immunoreactive proteins were detected using ECL (GE Healthcare, Little Chalfont, UK) and imaged using ImageQuant Las4010. Quantification of blots was performed using ImageJ imaging software.

## Accession numbers

Microarray data may be found at the Gene Expression Omnibus (GEO) under accession number GSE52141.

## Acknowledgements

We are grateful to G Damari and R Haffner-Krausz for transgenic mouse production and to O Higfa and Y Melamed for excellent mouse husbandry. Y Lotem, N Pencovich, O Ben-Ami, E Geron, B Shilo, E Schejter, A Sharp, R Straussman, and E Ariel contributed valuable know-how and protocols. K Buchman contributed to patent application PCT/IL2014/050064. M Pick, Y Groner and M de Brujin contributed insightful discussions, D Aronowitz performed English editing.

## Additional information

### Competing interests

LLI: Patent pending on the use of magnetic inorganic iron-based nanoparticles, PCT/IL2014/050064, EL: Patent pending on the use of magnetic inorganic iron-based nanoparticles, PCT/IL2014/050064, SM: Patent pending on the use of magnetic inorganic iron-based nanoparticles, PCT/IL2014/050064, J-PML: Patent pending on the use of magnetic inorganic iron-based nanoparticles, PCT/IL2014/050064. The other authors declare that no competing interests exist.

### Funding

| Funder | Grant reference number | Author |
| --- | --- | --- |
| Minerva Foundation | | Alexander Mildner |
| Israeli Science Foundation | | Steffen Jung, Eran Hornstein |
| Wolfson Family Charitable Trust | | Eran Hornstein |
| Carolito Stiftung | | Eran Hornstein |
| The Charlene Vener New Scientist Fund | | Eran Hornstein |
| Helen and Martin Kimmel Institute for Stem Cell Research | | Eran Hornstein |
| Julius and Ray Charlestein Foundation | | Eran Hornstein |
| Leir Charitable Foundation | | Steffen Jung |
| Deutsche Forschungsgemeinschaft | Research Unit 1336 | Steffen Jung |
| Fraida Foundation | | Eran Hornstein |
| ERC consolidator program | | Eran Hornstein |
| Kekst family institute for Medical Genetics | | Eran Hornstein |
| David and Fela Shapell Family center for Genetic disorders research | | Eran Hornstein |
| Crown Human Genome Center | | Eran Hornstein |
| Y. Leon Benoziyo Institute for Molecular Medicine | | Eran Hornstein |
| Women's Health Research Center | | Eran Hornstein |
| Estates of Fannie Sherr, of Lola Asseof and of Lilly Fulop | | Eran Hornstein |
| Dr. Sydney Brenner and friends | | Eran Hornstein |
| Helen and Milton A. Kimmelman Career Development Chair | | Eran Hornstein |

The funders had no role in study design, data collection and interpretation, or the decision to submit the work for publication.

### Author contributions

EC, conceived and led the project, contributed substantially to research conception and design, experimental activities and data analysis including establishing the miR-142 loss of function allele and developing indispensable methods, developed the interpretations presented and wrote the manuscript with

EH and approved the final version to be published; NR, led the revisions term of the work, contributed substantially to research design and data analysis with experimental activities *in-vivo* including molecular and cellular biology work and development of indispensable methods, critical contribution to drafting the article and approved the final version to be published; AM, contributed substantially to research conception and design, analysis and interpretation, experimental activities in bone marrow transplantation, competitive repopulation studies and flow cytometry, data analysis and developing of indispensable methods, critical contribution to drafting the article and approved the final version to be published; GB, generated the miR-142 null line, including collecting data and providing know-how and analysis of data which was critical for drafting the article and final approval of the version to be published; RP, performed Western blot analysis including collecting data and analysis by providing know-how that was critical for drafting the article and final approval of the version to be published; EM-R, helped performing research including collecting data and analysis and providing know-how, commented on analysis of data that was critical for drafting the article and final approval of the version to be published; YB, helped performing research including know how and highly relevant expertise, commented on analysis of data that was critical for drafting the article and final approval of the version to be published; GA, performed histopathology analysis including acquisition of data, analysis of data critical for drafting the article and final approval of the version to be published; IT, performed bioinformatics studies including acquisition of data, analysis of data critical interpretation of data for drafting the article and final approval of the version to be published; ZP, performed ImageStream flow cytometry including acquisition of data, analysis of data critical for drafting the article and final approval of the version to be published; LLI, EL, SM, J-PML, contributed unpublished essential reagents, critical input and resources, provided important intellectual content and final approval of the version to be published; SI, SJ, provided critical input, resources and scientific interpretations of data, provided critically important intellectual content and final approval of the version to be published; EH, conceived and supervised the study, developed the interpretations presented, wrote the manuscript and approved the final version to be published

### Ethics

Animal experimentation: This study was performed in strict accordance with the recommendations in the Guide for the Care and Use of Laboratory Animals of the National Institutes of Health. All of the animals were handled according to approved institutional animal care and use committee (IACUC) protocol of the Weizmann Instituter of Science. The protocol, entitled 'miR-142 in hematopoietic lineage development' was approved under Permit Numbers: 02930513-3 and 00350111-1. Every effort was made to minimize suffering.

## Additional files

### Supplementary file
• Supplementary file 1. Sequences of primers used in this study.

### Major dataset
The following dataset was generated:

| Author(s) | Year | Dataset title | Dataset ID and/or URL | Database, license, and accessibility information |
|---|---|---|---|---|
| Hornstein E, Chapnik E | 2013 | Comparison of gene expression profiles of miR-142−/− primary megakaryocytes and WT primary megakaryocytes | http://www.ncbi.nlm.nih.gov/geo/query/acc.cgi?acc=GSE52141 | Publicly available at NCBI Gene Expression Omnibus. |

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
