## [Decision Letter]

Thank you for sending your work entitled “miR-142 orchestrates a network of actin cytoskeleton cegulators during megakaryopoiesis” for consideration at *eLife*. Your article has been favorably evaluated by a Senior editor and 3 reviewers, one of whom is a member of our Board of Reviewing Editors.

The following individuals responsible for the peer review of your submission have agreed to reveal their identity: Elaine Fuchs (Reviewing editor); David Baltimore (peer reviewer). The other reviewer is remaining anonymous.

Your manuscript has now been seen by three experts in the field. As you can see from the reviews included at the end of this email, the comments are largely favorable and all agree that your work is potentially important and well-suited for publication in *eLife*. They indicate that for the most part, your evidence demonstrates that miR-142 is required for megakaryopoiesis and MK maturation into platelets, and that it functions by modeling cytoskeleton. That said, there are some significant concerns that will need to be addressed prior to moving forward with this paper.

Specifically:

1) Reviewers 1 and 2 raise issues regarding the extent to which you've documented that miR-142-3p is truly responsible for the phenotypes you describe. This seems especially critical given that the knockout was made through a gene trap strategy. Reviewer 1 points out that to argue the point that the observed megakaryopoiesis phenotype is solely due to miR-142‐3p loss, you should demonstrate that by introducing the 3p into the KO, you can fully restore functional MK maturation and platelet biogenesis. Along these lines, it also seems important to at least address whether miR-142b-5p on the other strand is expressed in your cells, whether the gene trap also disrupts its expression, and if so, whether it contributes to the observed megakaryopoiesis phenotype. Finally, reviewer 2 points out that even though you demonstrate the complete loss of miR-142, it is not clear whether the gene trap interferes with e.g. the noncoding RNA that is immediately downstream of the miR-142 coding region. While full rescues in vivo may be beyond the scope of the present study, you should at least confirm the phenotypes and address these points by re-introducing miR-142, 3p, and if relevant, 5p and Brzp1, into your sorted cells for in vitro rescue validations that the phenotypic consequences are indeed due to miR-142-3p loss.

2) Reviewers 2 and 3 persuasively point out the weakness in the data that allows you to conclude that up-regulation of actin-and cytoskeleton-related genes in miR-142 KO cells are responsible for their decreased platelet phenotype. Reviewer 3 suggests relatively straightforward in vitro experiments to bolster your data shown in Figure 5. Reviewer 2 asks for Western quantification in addition to qPCR and luciferase assays, and this seems reasonable. Again, with in vitro studies, which should not take so long to perform, further dissection could be done of how the derepression of actin targets contributes to the observed actin defects. Determining how the dosage affects your competing factors Grlf1/Cfl1 versus Was 1 again will help to discern miR142 functions in this regard. And explanations/discussions of whether the knockdown of your targets truly only show impact in KO and not WT cells are merited.

3) Reviewer 1 suggests that it will be valuable in aiding interpretation to analyze earlier stages of megakaryopoiesis where miR-142 KO MKs already show pronounced defects.

4) On the textual side, Reviewer 2 points out that miR-142 appears to widely express in hematopoietic lineages, and there are several key published papers that indicate important functions of miR-142 in HSC specification and differentiation. These should be discussed as should the issue of whether potential HSC defects could contribute to the observed defects in megakaryocytes. While carrying out experiments to explore this issue seem beyond what is required here, leaving open this possibility allows readers to consider the issue of whether this miRNA has megakaryocyte-specific functions or is more broadly involved in multiple cell types.

Reviewer #1:

This paper titled “miR-142 orchestrates a network of actin cytoskeleton regulators during megakaryopoiesis” presented solid evidence demonstrating that miR-142 is required for megakaryopoiesis and MK maturation into platelets. Several lines pointed to the mechanism that efficient megakaryopoiesis is achieved through regulating a cohort of actin cytoskeleton regulators. This finding is of significant importance. Most of the conclusions are drawn based on well-presented evidence. However, there are a few points as suggested below need to be further addressed before being considered further for publication in *eLife*: 1) miR-142b-3p resides on the (+) strand while there is another miRNA that resides within the same locus transcribed from the (-) strand; is miR-142b-5p expressed in this system? Does the gene trap also disrupt its expression? If so, does it contribute to the observed megakaryopoiesis phenotype?

2) The authors show that knocking down of miR-142's targets in miR-142-depleted MKs rescues PPF formation; how about earlier stages of megakaryopoiesis including MkP production and endomitosis where miR-142-/- MKs already reveals pronounced defects?

3) To help argue the point that the observed megakaryopoiesis phenotype is solely due to miR-142-3p loss, the authors should demonstrate that by supplementing miR-142-/- MK with miR-142-3p, they can fully restore functional MK maturation and platelet biogenesis.

Reviewer #2:

In this manuscript, Chapnik et al., describes differentiation defects in a miR-142 KO mouse during megakaryopoiesis. They demonstrated that miR-142, a miRNA highly expressed in hematopoietic lineages, is required for megakaryocyte maturation, in part through regulation of actin dynamics. The work represents an important contribution to understanding miRNA functions in the hematopoietic system and further supports miRNA networks as an important layer of regulation in diverse biological processes. I have several major concerns that should be addressed by the authors.

Major issues:

1) The KO mouse model was generated by a gene-trap strategy instead of a targeted approach. Although the authors demonstrated the complete loss of miR-142, it is not clear whether the insertion of the lacZ cassette interferes with other gene expression e.g. a noncoding RNA Bzrap1 is immediately downstream of the miR-142 coding region. The authors should confirm the phenotypes by re-introducing miR-142 to their sorted cells for at least in vitro validation.

2) miR-142 appears to widely express in hematopoietic lineages. Recent studies from zebrafish indicate important functions of miR-142 in HSC specification and differentiation (Lu X et al, Cell Research 2013; Nimmo et al., Dev Cell 2013). Although the current work focuses on megakaryopoiesis, the authors should at least discuss whether potential HSC defects could contribute to the observed defects in megakaryocytes. In particular, recent clonal analysis of HSCs indicates individual HSCs may contribute differently to distinct lineages (Lu R et al., Nature Biotech 2011). Since miR-142 is broadly expressed in almost all hematopoietic cells, these additional studies could provide critical insights for whether this miRNA has megakaryocyte-specific functions or more broadly involved in multiple cell types.

3) The authors have identified a number of miR-142 targets regulating the actin network through experimental and bioinformatic analyses. For target validation, they should provide Western quantification in addition to qPCR and luciferase assays. Furthermore, although multiple targets are all involved in regulating the actin dynamics, they have divergent functions as pointed out by the authors. For example, Cfl2, Twf1 and Grlf1 function in disassembly of actin filaments whereas Was1 promotes actin polymerization. Thus, it is critical to further dissect how the derepression of these targets contributes to the observed actin defects. Since the authors already knocked down individual targets with shRNA and, surprisingly, observed similar rescue between Grlf1/Cfl1 (disassembly factors) and Was1 (polymerization factor), they should address the issue how the dosage of these competing factors regulated by miR-142 plays important roles in the MK differentiation.

4) miRNA-mediated regulation is often subtle. However, Cfl1 mRNA is upregulated more than 8-fold in the KO (5D) despite the luc assay shows modest regulation (5F). Does the level of Cfl1 mRNA also indirectly regulated by the loss of miR-142?

5) The knockdown of individual targets by shRNA appears to be quite strong (Figure S4) e.g. 70-90% downregulation in the mRNA level. However, the upregulation of these mRNAs in miR-142 is generally modest except for Cfl1 (Figure 5). Thus, it's surprising that the knockdown of these targets only showed impact in the KO cells but not in the WT cells. I would assume the levels of these genes are probably similarly downregulated by these potent shRNAs in both WT and KO cells. Can the authors provide any further explanation?

Reviewer #3:

In this report, Chapnik et al. used miR-142 deleted mice to explore its role in hematopoiesis and found that it plays a role in platelet development. Accompanying this phenotype, they also reported that the megakaryocyte progenitor population was increased in the miR-142 null mice in the bone marrow and spleen. This effect was attributed to a cell intrinsic mechanism through the use of CD45.1/CD45.2 bone marrow chimera experiments. In complement with additional in vitro experiments, the authors conclude that miR-142 is important for maturation of megakaryocytes. The data presented in the manuscript supports this conclusion. Using genomic approaches, the authors identified a set of actin-related genes that were up-regulated upon miR-142 deletion. They suggest through 3'UTR reporter assays that these genes are direct targets of miR-142. Using RNAi techniques, the authors knocked down the expression of the actin-related genes in miR-142-/- cells and found that it rescued the ability of MKs to generate proplatelets using an in vitro proplatelet formation assay.

Overall, this manuscript is well written and their data in general supports their conclusion. The manuscript reveals a new function of miR-142 and puts miR-142 as a partial regulator of megakaryocytic maturation. However, their conclusion that up-regulation of actin- and cytoskeleton-related genes in miR-142-/- cells is responsible for their decreased platelet phenotype could be strengthened with additional experiments. The ultimate experiments would be to show that RNAi of these genes could increase platelet generation and decrease MkP population in the miR-142-/- mice. However, this requires additional long and laborious experiments. Thus, the authors could alternatively provide further in vitro evidence along with the experiment shown in Figure 5 (see comment #8 among the minor points). In summary, there are no major problems with this manuscript.

---

## [Author Response]

*1) Reviewers 1 and 2 raise issues regarding the extent to which you've documented that miR-142-3p is truly responsible for the phenotypes you describe. This seems especially critical given that the knockout was made through a gene trap strategy. Reviewer 1 points out that to argue the point that the observed megakaryopoiesis phenotype is solely due to miR-142‐3p loss, you should demonstrate that by introducing the 3p into the KO, you can fully restore functional MK maturation and platelet biogenesis*.

We have reintroduced miR-142-3p into the null model and ameliorated the megakaryocyte (MK) maturity defects, providing persuasive evidence that mature miR-142-3p is responsible for the observed phenotypes reported in our manuscript. The task was a major challenge because primary MKs are notoriously resilient to transfection. After being unsuccessful with commercially-available reagents, we tested a reagent that was recently developed in the labs of Shula Michaeli and Jean-Paul Lellouche at Bar Ilan University. This previously-unpublished reagent was critical for re-introduction of miR-142-3p into MKs and is a robust tool that will be probably utilized by many others in the field. It drove us to add the new tool into the Methods section and 4 additional authors that developed the reagent.

The cellular and molecular signature observed by re-gaining miR-142-3p function is shown in Figures 3 and 5. Figure 3 reports that the re-introduction of miR-142-3p using dsRNA mimetics was sufficient to restore WT PPF levels to miR-142-deficient differentiated MKs. In Figure 5, which focuses on targets of miR-142 upstream of a panel of cytoskeletal regulatory genes, we added a new panel 5E showing that dsRNA overexpression on the background of the genetic null allele recovers the molecular fingerprints of regulation. Thus, re-introduction of miR-142-3p using dsRNA mimetics was sufficient to restore Wasl and Cfl2 expression levels in miR-142-deficient differentiated MKs.

The relevant Results section reads: “Proplatelet formation (PPF) represents the final phase of MK maturation, culminating in platelet release into the bloodstream (35). To analyze whether miR-142 is involved in this process we performed an ex vivo PPF study on FL-derived MKs. Remarkably, we observed a striking 3-fold reduction in miR-142–/– MKs that were extending proplatelets, relative to control MKs (Figure 3). We next re-introduced miR-142-3p into differentiated MKs, using dsRNA mimetics (Figure 3—figure supplement 1). The introduction of miR-142-3p was sufficient to recapitulate WT PPF levels in cells that are genetically miR-142-deficient (Figure 3). Conversely, overexpression of miR-142-3p mimics in WT MKs did not yield any significant increase in PPF levels. Thus, miR-142-3p activity is essential for proper MK maturation and its loss results in defective platelet biogenesis.”

Importantly, the new sets of experiments allowed to make a new and important conclusion in the Results section: ”Furthermore, since re-introduction of miR-142-3p into miR-142-/- differentiated MKs restored functional identity, we conclude that there might be a continuous requirement for miR-142-3p activity to maintain MK maturity.” And in the Discussion: “…MKs require sustained miR-142-3p expression, as re-introduction of synthetic miR-142-3p mimetics, even onto differentiated MKs, was sufficient to restore functional maturity.”

*Along these lines, it also seems important to at least address whether miR-142b-5p on the other strand is expressed in your cells, whether the gene trap also disrupts its expression, and if so, whether it contributes to the observed megakaryopoiesis phenotype. Finally, reviewer 2 points out that even though you demonstrate the complete loss of miR-142, it is not clear whether the gene trap interferes with e.g. the noncoding RNA that is immediately downstream of the miR-142 coding region. While full rescues in vivo may be beyond the scope of the present study, you should at least confirm the phenotypes and address these points by re-introducing miR-142, 3p, and if relevant, 5p and Brzp1, into your sorted cells for in vitro rescue validations that the phenotypic consequences are indeed due to miR-142-3p loss*.

We added data that resolve concerns for both miR-142-5p and Brzp1 in a satisfactory manner. Our measurements reveal that miR-142-5p is expressed hundreds of folds less than miR-142-3p in megakarycyte and practically behaves like a typical ‘passenger’ strand and is discarded during the assembly process of the guide strand (miR-142-3p) into the RISC. See Figure 1. To the relevant text we added another reference that clearly views miR-142-5p as a passenger (non-functional) strand: “miR-142-3p is the guide strand from the miR-142 hairpin, whereas the sister ‘passenger’ miR-142-5p strand is expressed in negligible levels ((46) and Figure 1). The expression of both miR-142-3p and miR-142-5p was abolished in miR-142^–/–^ circulating blood cells as exemplified by quantitative real–time PCR (qPCR, Figure 1).”

Measurements of Bzrap1, a regulator of synaptic transmission, reveal that it unchanged in Megakaryocytes of miR-142 null mice. See Figure 2—figure supplement 1. The Results section reads: “We also confirmed that the expression of *Bzrap1*, a regulator of synaptic transmission (7; 43) positioned 3.5Kb downstream of miR-142, is unchanged in miR-142^–/–^ MKs (Figure 2—figure supplement 1).”

*2) Reviewers 2 and 3 persuasively point out the weakness in the data that allows you to conclude that up-regulation of actin-and cytoskeleton-related genes in miR-142 KO cells are responsible for their decreased platelet phenotype. Reviewer 3 suggests relatively straightforward in vitro experiments to bolster your data shown in*
Figure 5*. Reviewer 2 asks for Western quantification in addition to qPCR and luciferase assays, and this seems reasonable. Again, with in vitro studies, which should not take so long to perform, further dissection could be done of how the derepression of actin targets contributes to the observed actin defects*.

Measuring mRNA levels is an accepted practice in the miRNA field and some leading groups in the field suggest that in fact ‘mammalian microRNAs predominantly act to decrease target mRNA levels’ ([17] Nature. 466(7308):835-40). However, to comply with the reviewers’ request, we now provide on Figure 5 panels F-G assessment of target protein levels for representative miR-142-3p target genes, which substantiate our analysis. For the Western blot analysis, lysates from WT and miR-142–/– FL–derived MKs were subjected to SDS-polyacrylamide gel electrophoresis. WASL (F) and COFILIN-2 (G) were immunodetected and assessed by densitometry. The results section on page 10 reads: “Western blot analysis revealed upregulation of the protein products of Wasl and Cfl2 in miR-142–/– MKs, further substantiating them as bona fide miR-142-3p targets (Figure 5)”. We were unsuccessful in getting a purchased antibody against mouse Grlf1, another target of miR-142-3p, to work.

*Determining how the dosage affects your competing factors Grlf1/Cfl1 versus Was 1 again will help to discern miR142 functions in this regard*.

We performed additional experiments to further test the impact of different actin factors, which were described in the original submission. In a new Figure 6 we organized data supporting the claim that miR-142-3p targets a battery of actin cytoskeleton regulators to facilitate proplatelet formation. Panels 6B and 6C reveal the impact of miR-142^–/–^ MKs that were further transduced with shRNAs targeting individual (B), paired (C), or a combined set (B-C) of targets in restoring PPF levels.

As envisioned by the reviewers, more detailed analysis provided additional information. For example: Either Wasl or Cofilin knockdown restored PPF levels in miR-142-deficient MKs (i.e., recovered the wild-type phenotype). However, the knockdown of both targets simultaneously did not rescue the phenotype, presumably because of the opposing activity of Cofilin and Wasl on actin filamentous state. Another observation is that concomitant Cfl2 and Grlf1 knockdown, two proteins that are important for destabilizing actin polymers, enhanced PPF in miR-142 null MKs. The relevant Results section reads: “Additional pairwise target comparison revealed compound relationships between miR-142-3p targets (Figure 6; Figure 6—figure supplement 2). Remarkably, knockdown of both Cfl2 and Wasl, which carry out opposing functions within the actin regulatory network, did not yielded any significant increase in PPF levels in miR-142-deficient MKs (Figure 6). Conversely, concomitant Cfl2 and Grlf1 knockdown, two proteins that are important for destabilizing actin polymers, enhanced PPF in miR-142 null MKs (Figure 6). Lastly, a pool of shRNA against miR-142-3p targets was able to fully restore PPF capacity to WT levels (Figure 6). These data demonstrate that the expression of a set of actin cytoskeleton regulators should be tightly orchestrated by miR-142-3p in order to effectively promote platelet biogenesis.”

We further discuss how competing factors regulated by miR-142 may play important roles in the MK differentiation, while staying focused and sincere to the main claim of the work: “The actin cytoskeleton participates in a wide array of cellular functions, and the dynamic turnover between its F-actin and monomeric G-actin forms is regulated by a large number of actin-binding proteins. Within this cytoskeletal regulatory network, the set of genes targeted by miR-142-3p contains components with divergent functions. For example, Cfl2 and Twf1 participate in disassembly of actin filaments (1; 44). Likewise, Grlf1 is a GTPase activating protein that has been implicated in disruption of the organized actin cytoskeleton (39). Wasl, on the other hand, promotes actin polymerization by catalyzing filament branching together with the Arp2/3 complex (60). Thus, miR-142 deficiency destabilizes feedback loops required for actin filament homeostasis, stress fiber formation and actin remodeling. This in turn impairs proplatelet generation and plausibly other MK-intrinsic cellular functions.”

*And explanations/discussions of whether the knockdown of your targets truly only show impact in KO and not WT cells are merited*.

We discuss why shRNA knockdown of targets impacted the KO cells more than WT cells: “Repression of miR-142-3p actin-associated targets was found to be sufficient in restoring PPF levels in miR-142-deficient MKs. Interestingly, knocking down the same targets in WT MKs did not result in significant increase in PPF capacity over WT baseline levels. This might be related to a certain threshold in PPF levels that cannot be crossed, even when regulation of miR-142-3p is mimicked.”

*3) Reviewer 1 suggests that it will be valuable in aiding interpretation to analyze earlier stages of megakaryopoiesis where miR-142 KO MKs already show pronounced defects*.

In the revisions we measured several molecular markers of megakaryopoiesis to further characterize the affected steps in megakaryocyte differentiation in bipotent MK–erythroid precursors (PreMegEs) and downstream unipotent MK-progenitors (MkPs). We incorporated these new data into the work in Figure 2 and Results: “To gain insight into the impact of miR-142 nullification on early MK development, we performed a high–resolution flow cytometry assay for the characterization of myeloerythroid progenitors (54; 55)… Intriguingly, the expression levels of regulatory markers of MK development […namely Gata1 Gata2, Zfpm1, Klf1, Spi1, Fli1, Runx1, Tal1] remained largely unchanged in miR-142–/– PreMegEs and MkPs, relative to WT controls (Figure 2).”

In the Discussion we state: “miR-142 is dispensable for development of PreMegE, the bi-potent precursors of erythrocytes and MKs and the defects associated with miR-142 deficiency mostly relate to various facets of MK maturity, including increase in cell size, polyploidization and proplatelet elaboration.”

We further clarify in the revised manuscript that actin regulation is probably not the sole pathway downstream of miR-142 and other targets may be involved in regulating MkP proliferation and/or endomitosis. We also state: “It is also likely that additional miR-142 targets participate in the regulation of MK differentiation. For example, the over-representation of miR-142-deficient MkPs might result from a dysregulation of a distinct set of targets that are not necessarily related to actin regulation.”

*4) On the textual side, Reviewer 2 points out that miR-142 appears to widely express in hematopoietic lineages, and there are several key published papers that indicate important functions of miR-142 in HSC specification and differentiation. These should be discussed*.

We revised the Introduction and Discussion as requested and refer to other recently-published papers related to miR-142 activity in the hematopoietic lineage: “In the present work, we focused on miR-142, a hematopoietic-specific miRNA, which resides in a genomic locus that was previously associated with t(8;17) translocation in B–cell leukemia (15). Pioneering experimental evidence has suggested miR-142 involvement in lymphocyte differentiation (8) and recently, miR-142 was also shown to play a roles in the specification of definitive hemangioblasts (34; 46), and in lymphoid and myeloid lineages (3; 8; 15; 20; 28; 34; 46; 63; 66; 76). Furthermore, miR-142 is involved in compound immune response to North American eastern equine encephalitis virus (70) and our work uncovered a key role for miR-142 in the maintenance of CD4+ dendritic cells (42).” And in the Discussion section: “Hematopoietic–specific miR-142 emerges in recent years as a critical regulator of various blood and Lymph cell lineages (3; 8; 15; 20; 28; 34; 42; 46; 63; 66; 76)…”

*…as should the issue of whether potential HSC defects could contribute to the observed defects in megakaryocytes. While carrying out experiments to explore this issue seem beyond what is required here, leaving open this possibility allows readers to consider the issue of whether this miRNA has megakaryocyte-specific functions or is more broadly involved in multiple cell types*.

We performed analysis of earlier developmental stages and conclude that the phenotype emerges downstream of the PreMegE stage (bipotent erythroid-megakaryocyte progenitor) so it is a lineage-restricted phenotype. We come to this conclusion by in depth studies including in vivo competitive repopulation studies and in vitro-differentiated MK culture assays. Furthermore, the expression levels of several regulatory markers of MK development, including Gata1 Gata2, Zfpm1, Klf1, Spi1, Fli1, Runx1, Tal1, all remained unchanged in miR-142–/– PreMegEs. Even downstream at MK progenitors, most markers are unchanged and only Spi1, Fli1 are slightly upregulated (Figure 2). Therefore, we conclude that the phenotype which cannot be depicted at PreMegEs must emerge later in development.

We rephrased the text to better explain our interpretation that miR-142 holds megakaryocyte‐specific functions and is independently involved in other (equally intriguing) facets hematopoiesis and lymphopoiesis: “Our competitive repopulation studies and culture assays reveal that proper miR-142 function is essential, in a cell-intrinsic manner, for differentiation and maturation of MKs independently from additional important roles this miRNA has elsewhere in the hematopoietic system.”